# The long noncoding RNA lnc-FANCI-2 intrinsically restricts RAS signaling in human papillomavirus type 16-infected cervical cancer cells

Haibin Liu[1†‡], Lulu Yu[1†], Vladimir Majerciak[1], Thomas J Meyer[2], Ming Yi[3], Peter F Johnson[4], Maggie Cam[2], Douglas R Lowy[5], Zhi-Ming Zheng[1*]

[1]Tumor Virus RNA Biology Section, HIV Dynamics and Replication Program, CCR/NCI/NIH, Frederick, United States; [2]Collaborative Bioinformatics Resource, CCR/NCI/NIH, Bethesda, United States; [3]NCI RAS Initiative, Cancer Research Technology Program, Frederick National Laboratory for Cancer Research, Frederick, United States; [4]Mouse Cancer Genetics Program, CCR/NCI/NIH, Frederick, United States; [5]Laboratory of Cellular Oncology, CCR/NCI/NIH, Bethesda, United States

*For correspondence:
zhengt@exchange.nih.gov

†These authors contributed equally to this work

Present address: ‡Center for Emerging Infectious Diseases, Wuhan Institute of Virology, and Center for Biosafety Mega-Science, Chinese Academy of Sciences, Wuhan, Hubei, China

Competing interest: The authors declare that no competing interests exist.

## eLife Assessment

This study reports **important** new insights into the roles of a long noncoding RNA, lnc-FANCI-2, in the progression of cervical cancer induced by a type of human papillomavirus. Through a blend of cell biological, biochemical, and genetic analyses of RNA and protein expression, protein-protein interaction, cell signaling, and cell morphology, the authors provide **convincing** evidence that lnc-FANCI-2 affects cervical cancer outcome by regulating the RAS signaling pathway. These findings will be of interest to scientists in the fields of cervical cancer, long noncoding RNA, and cell signaling.

**Abstract** Increased expression of lnc-FANCI-2, a newly discovered long noncoding RNA, is associated with cervical lesion progression from cervical intraepithelial neoplasia stage 1 (CIN1, low grade), CIN2–3 (high grade), to cervical cancer. Viral oncoprotein E7 of high-risk human papillomaviruses (HR-HPVs) and host transcription factor YY1 are two major factors promoting lnc-FANCI-2 expression. Using CRISPR-Cas9 technology, we knocked out the expression of *lnc-FANCI-2* in the HPV16-positive cervical cancer cell line, CaSki cells. The selected knockout (KO) single-cell clones displayed altered cell morphology and proliferation with changes of cellular soluble receptors, but normal HPV16 E6 and E7 expression. Relative to the parental cells, lnc-FANCI-2 KO cells exhibited significantly increased RAS signaling and epithelial-mesenchymal transition, but decreased response to IFN signaling, along with increased p-Akt and p-Erk1/2 (two RAS signaling effectors), IGFBP3, MCAM, VIM, and CCND2 (cyclin D2) and decreased expression of RAC3. Lnc-FANCI-2 in CaSki interacts with cellular proteins H13, HNRH1, K1H1, MAP4K4, and RNPS1. MAP4K4 knockdown led to enhance the expression of p-Erk1/2 and p-Akt. High lnc-FANCI-2 and low MCAM levels in cervical cancer tissues were found to be associated with patients' survival. A key function of lnc-FANCI-2 intrinsically regulates RAS signaling to impact cervical lesion progression and cervical cancer prognosis.

## Introduction

Cervical cancer is a leading cause of mortality in women worldwide. It is the second most diagnosed cancer among women, with an estimated 661,000 new cases worldwide each year, and the third most frequent cause of cancer-related death, accounting for 348,000 deaths annually (*Bray et al., 2024*). Persistent infections in the cervical epithelium with the cancer-associated alpha human papillomaviruses (high-risk HPVs [HR-HPVs]), in particular HPV16 and HPV18 with the sustained expression of viral oncoproteins, E6 and E7, contribute to squamous epithelial cell tumorigenesis (*Bosch et al., 2002*; *Munoz et al., 2003*; *Walboomers et al., 1999*). Approximately 31.3% of high-grade neoplastic cervical lesions progress to invasive cervical cancer (ICC) in 30 years (*McCredie et al., 2008*), while most lesions resolve spontaneously. The progression to cervical cancer typically takes 15–20 years after initial infection (https://www.who.int/news-room/fact-sheets/detail/cervical-cancer). These observations suggest that active host defense systems play important roles in preventing malignant transformation of neoplastic cervical lesions, and additional factors besides E6 and E7 are necessary for cervical cancer development. Indeed, one tumor suppressor network, the Fanconi anemia (FA) pathway (*Niraj et al., 2019*), can be activated by HPV-induced DNA damage. FA proteins, including FANCA, FANCD2, and FANCI, are elevated upon HPV infection (*Hoskins et al., 2009*; *Liu et al., 2021*; *Santegoets et al., 2012*), and the activated FA pathway restricts HPV replication and guards host genome stability (*Hoskins et al., 2012*). FANCD2 mutation stimulates HPV16 and HPV31 genome amplification and also promotes cervical and vaginal cancer development in HPV16 E7 transgenic mouse (*Park et al., 2010*; *Park et al., 2014*).

HPV-positive cervical cancer tissues in general exhibit wild-type (WT) p53, WT K-RAS, and no overexpression of RAS genes (*Hietanen et al., 2000*; *Tommasino et al., 2003*; *Prior et al., 2012*; *Fernández-Medarde and Santos, 2011*). Viral oncoprotein E7 plays a major role in the immortalization of primary epithelial cells mainly by inactivation of pRb family of proteins (*Roman and Munger, 2013*), whereas E6 mediates degradation of tumor suppressor p53 and activation of human telomerase reverse transcriptase transcription (*Vande Pol and Klingelhutz, 2013*; *Liu et al., 2009*; *Klingelhutz et al., 1996*). However, high-risk E6 and E7 are necessary but not sufficient to transform fully the immortalized cells into malignant cells (*Storey et al., 1988*; *Barbosa and Schlegel, 1989*; *Hawley-Nelson et al., 1989*; *Münger et al., 1989*; *Halbert et al., 1991*), and additional oncogenic stress is needed. For example, the HRAS$^{G12V}$, a constitutively activated RAS GTPase mutation, triggers malignant transformation of E6/E7-expressing keratinocytes (*Phelps et al., 1988*; *DiPaolo et al., 1989*; *Schreiber et al., 2004*; *Narisawa-Saito et al., 2008*). Normal RAS regulates cell proliferation, cell differentiation, and cell adhesion through cellular signal transduction from growth factor receptors such as insulin-like growth factors (IGFs) and insulin-like growth factor binding proteins (IGFBP1–6) (*Conover et al., 2000*; *Allard and Duan, 2018*; *Bach, 2018*). Thus, overactive RAS signaling might facilitate transformation of E6/E7-expressing keratinocytes. In transgenic mice, HPV16 E6 and E7 can promote the development of spontaneous epithelial skin tumors, but not spontaneous tumors of the reproductive tract (*Song et al., 1999*; *Herber et al., 1996*; *Song et al., 2000*; *Riley et al., 2003*). Prolonged estrogen treatment is required (*Brake and Lambert, 2005*). Estrogen receptor (ER) activation stimulates the mitogen-activated protein kinase (MAPK/Erk) and phosphoinositide 3-kinase (PI3K/Akt) pathways, both of which are mediated by RAS (*Kato et al., 1995*). However, the molecular mechanism underlying estrogen-induced cervical cancer in E6 and E7 transgene mice is not fully understood.

Long noncoding RNAs (lncRNAs) are RNAs over 200 nucleotides (nt) in length lacking a coding capacity for translation into functional proteins. Although several lncRNAs have been proposed to function in diverse biological processes, evidence is still lacking to support the functionality of the majority of lncRNAs (*Statello et al., 2021*; *Liu and Zheng, 2022*).

We recently reported an increased level of lnc-FANCI-2 expression in high-risk HPV-positive cervical intraepithelial neoplasia (CIN) and ICC tissues (*Liu et al., 2021*). At least 14 RNA isoforms of lnc-FANCI-2 are expressed by usage of two alternative promoters, alternative RNA splicing, and alternative selection of two polyadenylation sites from the genomic locus adjacent to the FANCI on Chr15. Ironically, the lnc-FANCI-2 was wrongly annotated by NCBI as a *MIR9-3HG* in the recently updated RefSeq database (https://www.ncbi.nlm.nih.gov/refseq/) and two alternative lnc-FANCI-2 promoters identified for expression of lnc-FANCI-2 are ~10 kb downstream of the non-expressible *MIR9-3* gene (*Liu et al., 2021*). Both lnc-FANCI-2 and FANCI are upregulated simultaneously in neoplastic

cervical lesions and cervical cancer by HR-HPV infections. E6, and in particular, E7 is responsible for the enhanced expression of lnc-FANCI-2, the transcription of which is also regulated by YY1 (*Liu et al., 2021*). HPV infection increases YY1 levels but decreases the expression of p53-dependent miR-29a which targets the YY1 3' UTR. Viral E7 interacts with YY1 and facilitates YY1 transactivation of lnc-FANCI-2 promoter (*Liu et al., 2021*). In situ hybridization has revealed that lnc-FANCI-2 is preferentially cytoplasmic, but its function in HPV-infected cells has not been characterized. In this report, we demonstrate that lnc-FANCI-2 in HPV16-infected cells controls RAS signaling by interaction with MAP4K4 and other RNA-binding proteins. Ablation of lnc-FANCI-2 in the cells promotes RAS signaling and phosphorylation of Akt and Erk. High levels of lnc-FANCI-2 and low levels of MCAM expression in cervical cancer patients correlate with improved survival, indicating that lnc-FANCI-2 plays a critical role in regulating RAS signaling to affect cervical cancer progression and patient outcomes.

## Results

### Differential expression of lnc-FANCI-2 in cervical cancer tissues and its derived cell lines

As lnc-FANCI-2 expression is upregulated along with cervical lesion progression by HR-HPV infections (*Liu et al., 2021*), we further verified the increased expression of lnc-FANCI-2 in cervical cancer tissues by RNAscope single-molecule RNA in situ hybridization (RNA-ISH) using an lnc-FANCI-2 antisense probe spanning over the major isoform of lnc-FANCI-2 nt 359–1713 region (GenBank MT669800.1). We found both cytoplasmic and nuclear locations of the increased lnc-FANCI-2 ISH signals within the tumor nest in HPV16-infected cervical squamous cell carcinoma tissues, but not much so in its adjacent normal tissue areas (*Figure 1A/B*). The increased expression of lnc-FANCI-2 became obvious in human foreskin keratinocytes with HPV16 or HPV18 infection when compared to the HFK cells without HPV infection (*Figure 1C*).

However, we subsequently demonstrated the high expression of lnc-FANCI-2 in two HPV16-infected cervical cancer cell lines (SiHa and CaSki) and two HPV16-infected low-grade cervical lesion-derived cervical keratinocyte subclone lines (W12 20,861 and W12 20,863 cells), but not in two HPV18-infected cervical cancer cell lines (HeLa and C4II cells), nor other HPV-negative cell lines of HCT116 (colorectal cancer cells), BCBL-1 (body cavity B lymphoma cells), HEK293 (Ad5 E1/E2-immortalized human kidney cells), and HaCaT (spontaneously immortalized human epidermal cells) (*Figure 2A*). Interestingly, the high expression of lnc-FANCI-2 was observed in an HPV-negative cervical cancer cell line C33A cells with mutations of both p53 and RB genes (*Scheffner et al., 1991*; *Figure 2A*). Northern blot confirmed the increased expression of lnc-FANCI-2 in C33A cells but no expression in HeLa cells (*Figure 2B*).

Subcellular distribution of lnc-FANCI-2 was characterized by cell fractionation and western blot assays using nuclear SRSF3 and cytoplasmic GAPDH protein as an indication of fractionation efficiency (*Figure 2C*). By RT-qPCR analysis of the fractionated total RNA, we demonstrated that lnc-FANCI-2 RNA is mainly cytoplasmic in HPV16-positive CaSki cells, but nuclear in HPV16-positive SiHa and HPV-negative C33A cells (*Figure 2C*). The differential subcellular distributions of lnc-FANCI-2 RNA from CaSki to SiHa and C33A cells (*Figure 2D*) were further confirmed by RNAscope RNA-ISH using the lnc-FANCI-2 antisense probe described above.

### lnc-FANCI-2 regulates proliferation of HPV-transformed cervical cancer cells

lnc-FANCI-2 RNA is transcribed mainly from a proximal promoter TSS2, but also from an alternative minor, distal promoter TSS1. Two highly conserved YY1 binding motifs upstream of the TSS2 are essential for the TSS2 transcriptional activity (*Liu et al., 2021*; *Figure 3A*). To elucidate the function of lnc-FANCI-2 in high-risk HPV-infected cervical cancer cells, we knocked out lnc-FANCI-2 expression in HPV16-positive CaSki cells using CRISPR/Cas9 deletion of either a 3 kb promoter region encompassing both TSS1 and TSS2 or an 86 bp region containing two YY1-binding motifs in the TSS2 promoter (*Figure 3A*). Two tested guide RNAs (gRNAs) with high KO efficiency were selected and cloned into a modified CRISPR/Cas9 expression vector to express both 5'-specific gRNA and 3'-specific gRNA simultaneously for efficient genome editing (*BeltCappellino et al., 2019*). CaSki cells stably transfected with the dual gRNA expression vector were generated. Through a serial dilution,

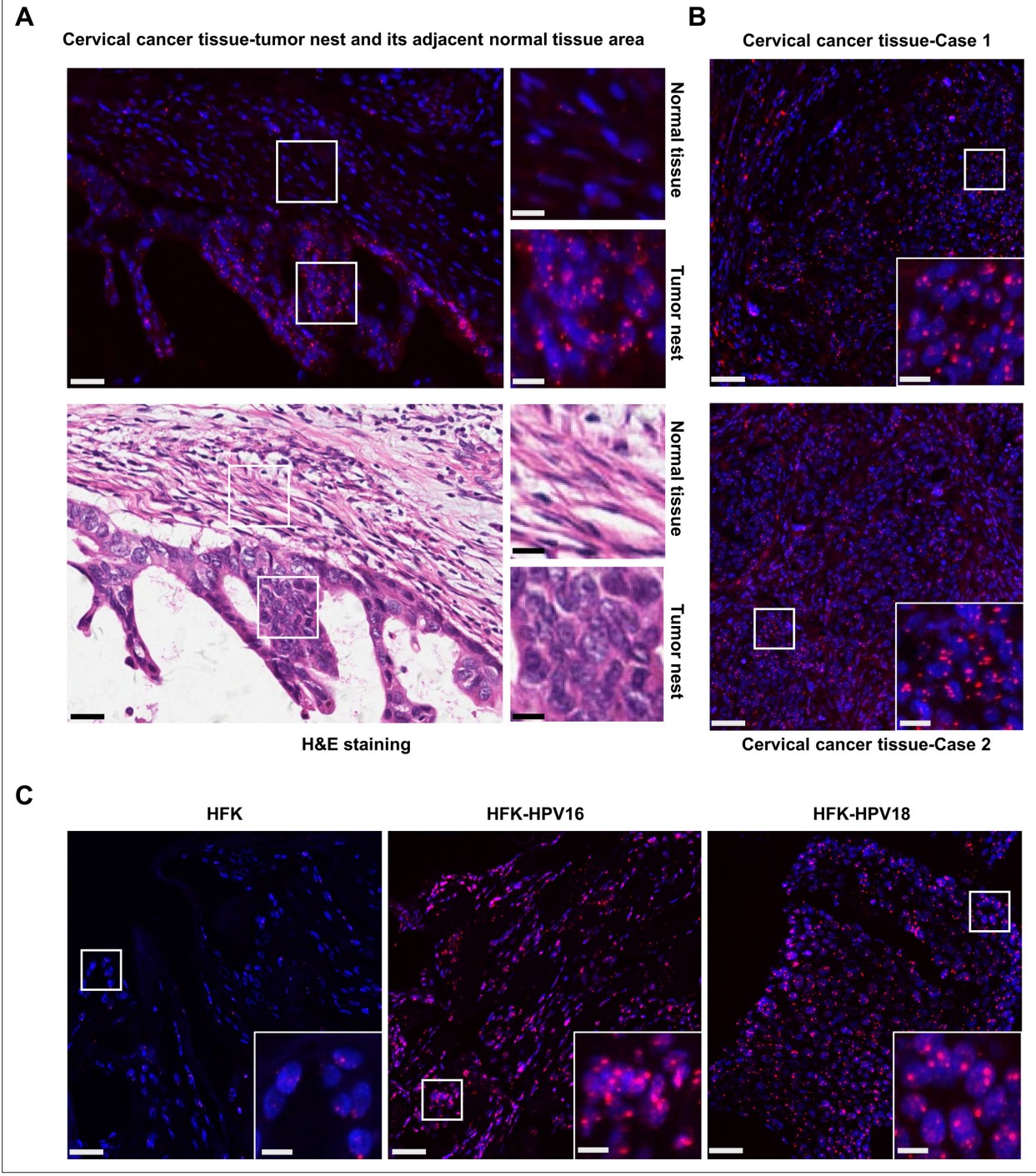

**Figure 1.** Increased expression of lnc-FANCI-2 in HPV16-infected cervical squamous cell carcinoma tissues and HPV16- and HPV18-infected raft culture tissues. (**A** and **B**) Expression of lnc-FANCI-2 (red) in HPV16[+] cervical squamous cell carcinoma tissues was examined by RNAscope RNA in situ hybridization (RNA-ISH) analysis. Nuclei were stained with DAPI (blue). The corresponding regions (**A**) in H&E staining are shown for the adjacent region from the tumor nest and histotype. (**C**) Human foreskin keratinocyte (HFK)-derived raft cultures without (HFK) or with HPV16 or HPV18 infections (HFK-HPV16 or HFK-HPV18) were examined at day 10 for lnc-FANCI-2 RNA by RNAscope RNA-ISH analysis. Scale bars: 25 µm in the original figures and 10 µm in the zoomed insets.

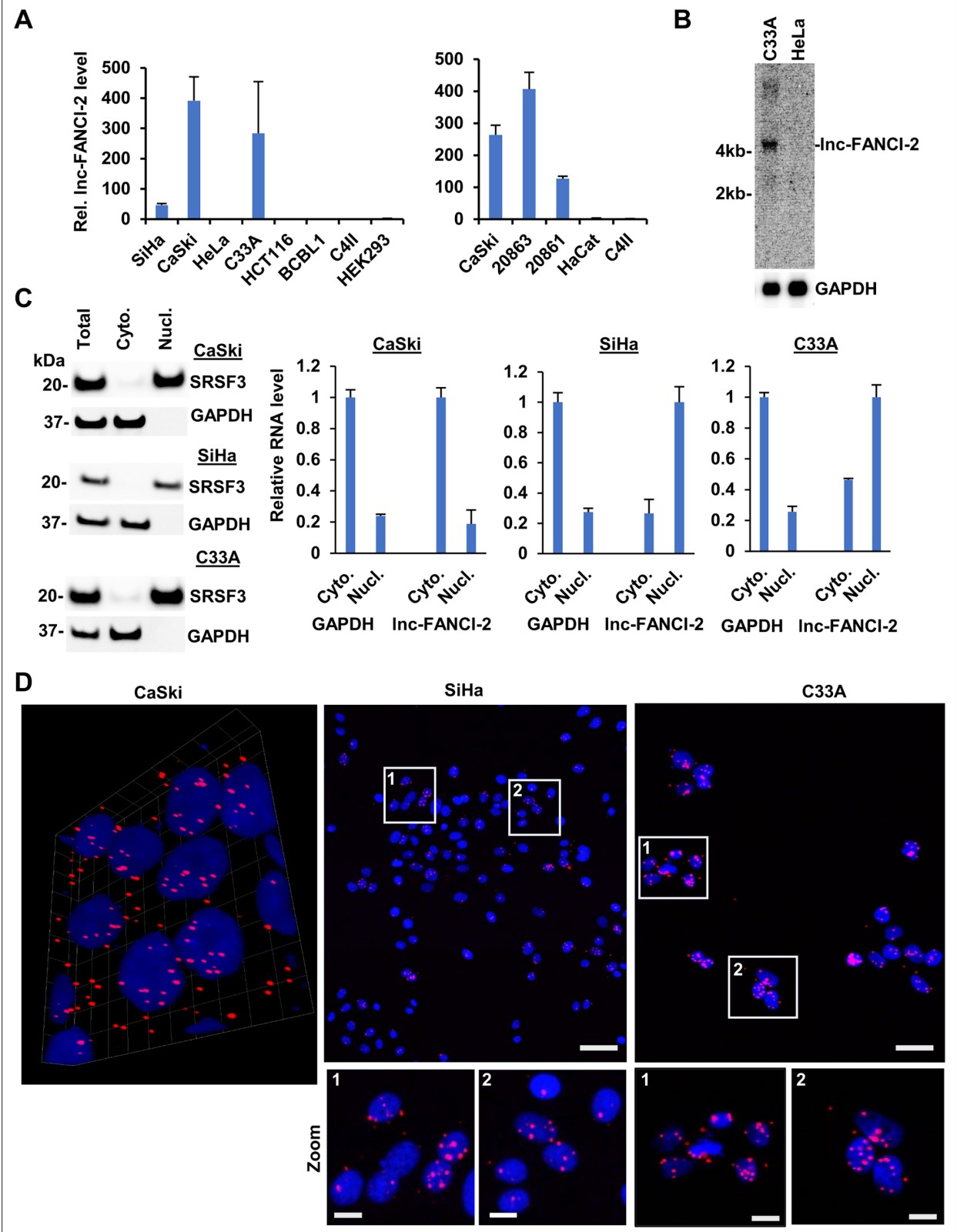

**Figure 2.** Selective expression of lnc-FANCI-2 RNA in cervical cancer cell lines. (**A**) RT-qPCR detection of lnc-FANCI-2 in HPV16+ cervical cancer cell lines SiHa, CaSki, and W12 20861 (integrated HPV16) and 20863 (episomal HPV16), HPV18+ cervical cancer cell lines HeLa and C4II, and HPV- cell lines C33A (cervical cancer cells with mutations of p53 and pRb), HCT116 (colorectal cancer cells), BCBL-1 (body cavity B lymphoma cells), HEK293 (Ad5 E1/E2-immortalized human kidney cells), and HaCaT (spontaneously immortalized human epidermal cells) in triplicates. (**B**) HeLa cells express no lnc-FANCI-2

*Figure 2 continued on next page*

*Figure 2 continued*

when compared with C33A cells by northern blot. (**C**) lnc-FANCI-2 is mainly cytoplasmic in CaSki but nuclear in SiHa and C33A cells. Cytoplasmic and nuclear fractionation efficiency was blotted for nuclear SRSF3 (serine- and arginine-rich splicing factor 3) and cytoplasmic GAPDH. Total fractionated cytoplasmic and nuclear RNAs were quantified for lnc-FANCI-2 by RT-qPCR in triplicates, with GAPDH RNA serving as an internal control for RNA fractionation efficiency. (**D**) Subcellular lnc-FANCI-2 (red) localization in CaSki, SiHa, and C33A cells determined by RNAscope RNA in situ hybridization (RNA-ISH) analysis. Nuclei were stained with DAPI (blue). Scale bars: 25 µm in the top and 10 µm in the zoom.

The online version of this article includes the following source data for figure 2:

**Source data 1.** Northern blot and western blot data for *Figure 2*.

**Source data 2.** Northern blot and western blot data for *Figure 2*.

several single-cell clones were isolated and verified for homozygous deletion by PCR screening. PCR genotyping indicated successful homozygous deletion of the promoter or YY1-binding motifs, respectively (*Figure 3B*).

Loss of lnc-FANCI-2 expression in the single-cell clones was examined by RT-qPCR (*Figure 3C*). When compared to all control clones transfected and selected from an empty vector containing no gRNA, all single-cell clones with deletion of YY1-binding motifs (ΔYY1) showed 70–80% reduction of lnc-FANCI-2 expression, whereas the single-cell clones with deletion of both TSS1 and TSS2 promoters (ΔPr) showed >90% reduction (*Figure 3C*). Northern blot analysis confirmed the decrease in levels of the two major lnc-FANCI-2 isoforms (*Figure 3D*), with an abundant 4 kb lnc-FANCI-2 RNA derived from a distal polyadenylation site pA2 and a less abundant 2 kb of lnc-FANCI-2 RNA derived from a proximal polyadenylation site pA1 (*Liu et al., 2021*; *Figure 3A and D*). Reduced lnc-FANCI-2 RNA expression was also confirmed by RNAscope RNA-ISH (*Figure 3E and F*). The parental (WT) CaSki cells expressed ~10 copies of lnc-FANCI-2 RNA per cell (*Liu et al., 2021*), whereas the copy number of lnc-FANCI-2 RNA in the ΔYY1-D5 cells dropped to ~5 copies and to ~2 copies per cell in the ΔPr-A9 cells (each averaged from 200 cells, *Figure 3F*).

Although the WT CaSki cells preferentially grow as distinct cluster cell islands, the lnc-FANCI-2 KO cells displayed a dispersed cell growth pattern, and some cells exhibited an irregular or spindle-like cell morphology (*Figure 3—figure supplement 1*). The KO cells also grew slower than the parental CaSki cells (*Figure 3G*), formed fewer colonies with reduced size (*Figure 3H*), and showed decreased migration capacities (*Figure 3I*), with ΔPr-A9 cells having a more severe phenotype than ΔYY1-D5 cells. Subsequently, the ΔPr-A9 and ΔPr-B3 cells were further examined for their HPV16 E6 and E7 expression and the ΔPr-A9 for cell senescence. We found both ΔPr-A9 and ΔPr-B3 cells, when compared with parental WT CaSki cells, exhibited some small, variable changes in the expression of E6, E7, p53 (E6 downstream target) and E2F1 (E7 downstream target) proteins (*Figure 3E and I*, *Figure 3—figure supplement 2A*), most likely from sampling in the assays, but the ΔPr-A9 cells displayed significant increase in cell senescence (*Figure 3G and I*, *Figure 3—figure supplement 2B*). Altogether, the data indicate that lnc-FANCI-2 RNA is important for cell growth, colony formation, and cell migration.

## lnc-FANCI-2 regulates the expression and secretion of cell soluble receptors

Considering that the altered expression of cell membrane proteins and secreted factors in the lnc-FANCI-2 KO cells might contribute to the observed different cell morphology and growth properties, we performed a Proteome Profiler Human sReceptor Array analysis to examine possible changes in the expression of 105 well-characterized soluble protein receptors being involved in cell signaling using total cell lysates and cell culture supernatants from ΔPr-A9 and WT CaSki cells.

Using a 30% cutoff (FC -/+ 0.3), we identified from the ΔPr-A9 cell lysate nine proteins with increased expression compared to the WT CaSki cells, including PODXL2, ECM1, NECTIN2, MCAM, ADAM9, ADAM10, CDH5, ITGA5, and NOTCH1, and six proteins with decreased expression, including ITGB6, CDH13, LGALS3BP, TIMP2, ADAM8, and SCARF2 (*Figure 4A and B*). We also found from the ΔPr-A9 cell culture supernatant five proteins with increased expression, including ADAM9, NECTIN2, ADAM10, CDH5, and ECM1, and the decreased expression of four proteins, including CRELD2, SDC1, SDC4, and TIMP2 (*Figure 4A and B*). By immunoblot assays, we verified selectively the increase in PODXL2, MCAM, and ECM1 and the decrease in ADAM8 and TIMP2 from the ΔPr-A9 cell lysates and the increase in ECM1 and the decrease in TIMP2 in the ΔPr-A9 cell culture supernatant (*Figure 4C*).

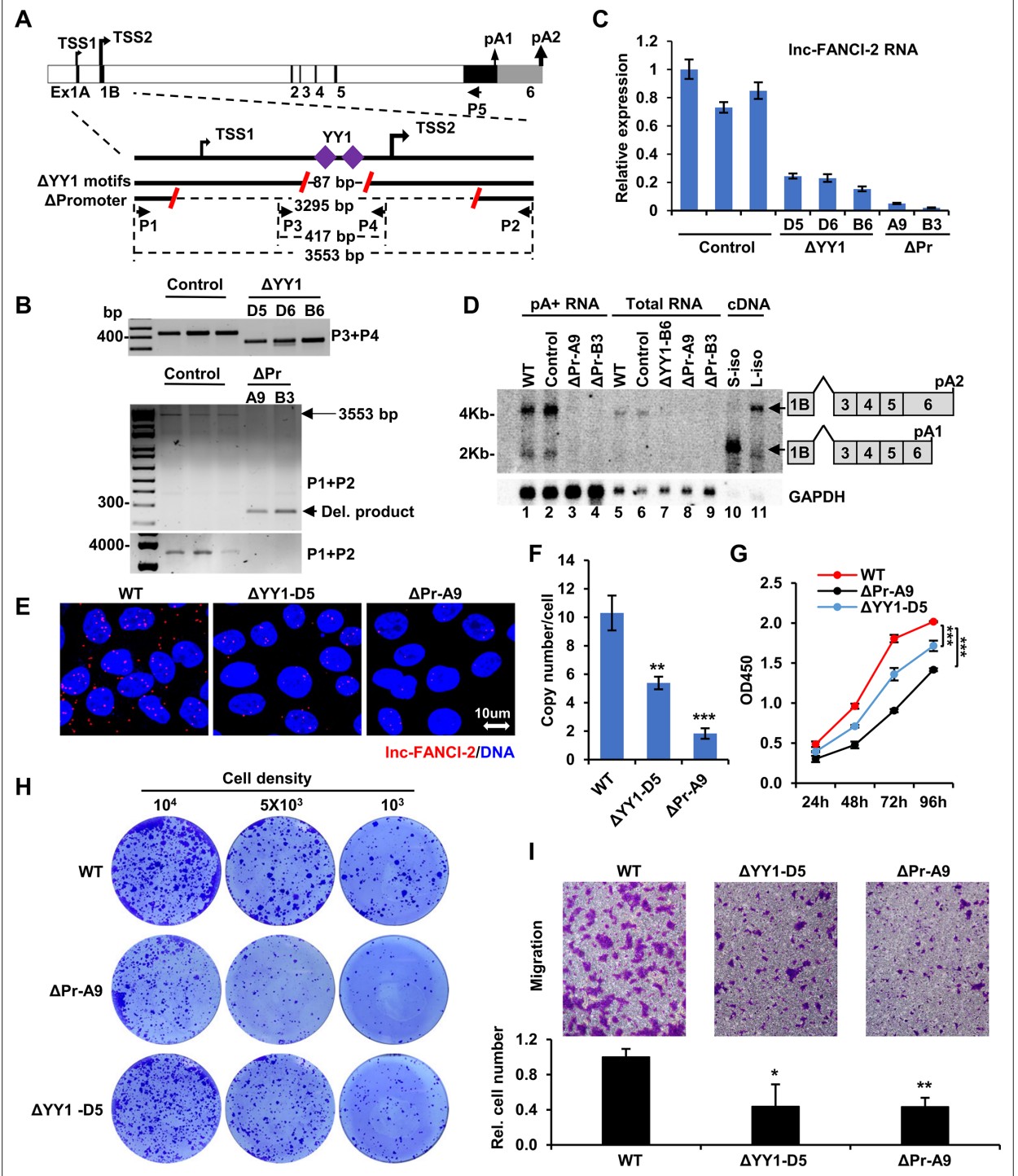

**Figure 3.** Knockout (KO) of lnc-FANCI-2 in CaSki cells affects cell proliferation, colony formation, and migration. (**A**) Diagram and KO strategies of lnc-FANCI-2 gene. On the figure top is the lnc-FANCI-2 gene structure and its alternative transcription start sites (TSS) and polyadenylation sites (pA). TSS2 or pA2 with a heavier arrow is predominately used for lnc-FANCI-2 expression. The lower figure part shows KO strategies. Red slashes represent guide RNA (gRNA)-targeted sites to create genomic DNA deletions by CRISPR-Cas9 technology. Deletion of two YY1-binding motifs (ΔYY1) led to a deletion of 86 bp DNA fragment. Deletion of lnc-FANCI-2 promoter (ΔPr) led to deleting an ~3.3 kb promoter region. Primers (P1–P4) used for PCR screening were shown as black arrows. (**B**) PCR screening of single-cell clones with homozygous lnc-FANCI-2 KO, with indicated primer sets diagrammed (**A**). Single-cell clones selected from the cells transfected with an empty vector served as control (Ctrl) cells. (**C**) Evaluation of lnc-FANCI-2 KO efficiency in the selected single-cell clones (**B**) by RT-qPCR. (**D**) Northern blot validation of lnc-FANCI-2 KO efficiency from the individual single-cell clones or the parental (wild-type [WT]) CaSki cells on polyA+ RNA (enriched from 100 µg of total cell RNA, lanes 1–4) or 10 µg of total cell RNA (lanes 5–9). Total cell

*Figure 3 continued on next page*

*Figure 3 continued*

RNA (2 μg) of HEK293T cells with ectopic expression of two isoforms (short, S or long, L) of lnc-FANCI-2 cDNA served as a control. Antisense oligo probe P5 (**A**) labeled with $^{32}$P was used for hybridization. GAPDH RNA served as a loading control and hybridized with a GAPDH-specific oligo probe. (**E–F**) RNA in situ hybridization (RNA-ISH) validation of lnc-FANCI-2 KO efficiency by RNAscope technology. Two single-cell clones were examined with red color for lnc-FANCI-2 and blue color for the nucleus (**E**). Bar graphs show the copy number of lnc-FANCI-2 per cell in the WT, ΔYY1-D5, or ΔPr-A9 CaSki cells (each averaged from 200 cells) (**F**). (**G–I**) Cell proliferation, colony formation, and migration of WT, ΔYY1-D5, or ΔPr-A9 cells show requirements of lnc-FANCI-2 expression for cell proliferation, colony formation, and cell invasion of CaSki cells. Representative pictures are shown from three separate experiments of cell proliferation by CCK8 assay (**G**), cell colony formation under 1% methylcellulose solution (**H**), and cell migration by transwell assay (**I**) of WT, ΔYY1-D5, or ΔPr-A9 cells. Cell colonies were stained with crystal violet as described in the procedure. Numbers of the invaded cells that crossed the transwell membrane were quantified and shown as bar graphs (**I**). *, p<0.05; **, p<0.01; ***, p<0.001 by two-tailed Student's t-test.

The online version of this article includes the following source data and figure supplement(s) for figure 3:

**Source data 1.** PCR screening and northern blot data of lnc-FANCI-2 KO single cell clones for *Figure 3*.

**Source data 2.** PCR screening and northern blot data of lnc-FANCI-2 KO single cell clones for *Figure 3*.

**Figure supplement 1.** lnc-FANCI-2 knockout (KO) on characteristic CaSki cell growth behavior and morphology imaged at 24 hr after spreading.

**Figure supplement 2.** lnc-FANCI-2 knockout (KO) on expression of viral oncoproteins and cell senescence.

**Figure supplement 2—source data 1.** lnc-FANCI-2 KO in CaSki cells and the expression of HPV16 E6 and E7 and their downstream targets.

**Figure supplement 2—source data 2.** lnc-FANCI-2 KO in CaSki cells and the expression of HPV16 E6 and E7 and their downstream targets.

As ADAM8 is proteolytically processed into two protein isoforms for cell adhesion (*Schlomann et al., 2002*) and migration (*Conrad et al., 2022*), we confirmed by immunoblot the decreased expression of all three sizes of ADAM8 protein in the ΔPr-A9 cell lysate over the WT CaSki cells (*Figure 4C*).

## lnc-FANCI-2 regulates the expression of genes involved in RAS signaling

We next conducted genome-wide RNA-seq analyses of WT and lnc-FANCI-2 KO cells (4 samples/group) to determine the transcriptomic consequences of lnc-FANCI-2 deficiency. We obtained ~120 million mappable RNA reads to the human reference genome hg38 from each sample. Analysis of RNA-seq reads-coverage map by Integrative Genomics Viewer (IGV) showed ~90% reduction of lnc-FANCI-2 expression in ΔPr-A9 cells and ~72% decrease in ΔYY1-D5 cells compared to the WT CaSki cells (*Figure 5A*), consistent with the results shown in *Figure 3C–F*.

Hierarchical clustering analysis showed more global transcriptional similarity between ΔYY1-D5 and ΔPr-A9 cells (*Figure 5B*). By applying a threshold of fold change (FC)≥1.8 or FC ≤–1.8 with FDR ≤0.01, we found 1230 genes in ΔPr-A9 and 797 genes in ΔYY1-D5 cells that were significantly differentially expressed relative to the WT CaSki cells expressing ~15,890 genes. Nine (*Italic*) of 15 (60%) soluble receptor proteins in cell lysates (PODXL2, ECM1, *NECTIN2, MCAM, ADAM9*, CDH5, ADAM10, *ITGA5*, *NOTCH1*, SCARF2, ADAM8, *TIMP2*, *LGALS3BP*, *CDH13*, and *ITGB6*) exhibited consistent changes in expression by both RNA-seq and protein array assays. The most significantly affected genes are shown in Volcano plots (*Figure 5C* and *Supplementary file 1*). Among these, 211 were upregulated and 189 were downregulated in both ΔPr-A9 and Δ YY1-D5 cells (*Figure 5D* and *Supplementary file 2*). By applying more stringent criteria with FPKM ≥7 in at least one of four RNA-seq samples as a cutoff, we profiled the genes with large expression differences in both ΔPr-A9 and ΔYY1-D5 cells. The results displayed in the heatmap of *Figure 5E* included 52 upregulated and 47 downregulated genes, excluding lnc-FANCI-2. Subsequently, we selectively verified by RT-PCR the decreased expression of ITGB6, NLRP2, PLAC8, PSG4, PSMB9, and SERPINB1 and the increased expression of CFH, CNTN5, EMP3, IGFBP3, PTGS2, and SERPINB2 in ΔPr-A9 cells (*Figure 5F*), as well as the increased expression of IGFBP3 protein, a RAS signaling driver (*Conover et al., 2000*; *Allard and Duan, 2018*; *Bach, 2018*), in ΔPr-A9 cells by immunoblot (*Figure 5G*).

By performing the gene set enrichment analysis (GSEA) on the Hallmark gene sets, which provide more refined and concise inputs for GSEA (*Liberzon et al., 2015*), we found the most significantly upregulated pathways in both ΔPr-A9 and ΔYY1-D5 cells, when compared with the WT CaSki cells, were KRAS signaling and epithelial-mesenchymal transition (EMT). The most significantly downregulated pathways in both ΔPr-A9 and ΔYY1-D5 cells over the WT CaSki cells were interferon gamma (IFN-γ) and interferon alpha (IFN-α) responses (*Figure 6A*). GSEA plots for ΔPr-A9 show 29 of 111 genes, including IGFBP3 (*Conover et al., 2000*; *Allard and Duan, 2018*; *Bach, 2018*), in RAS signaling

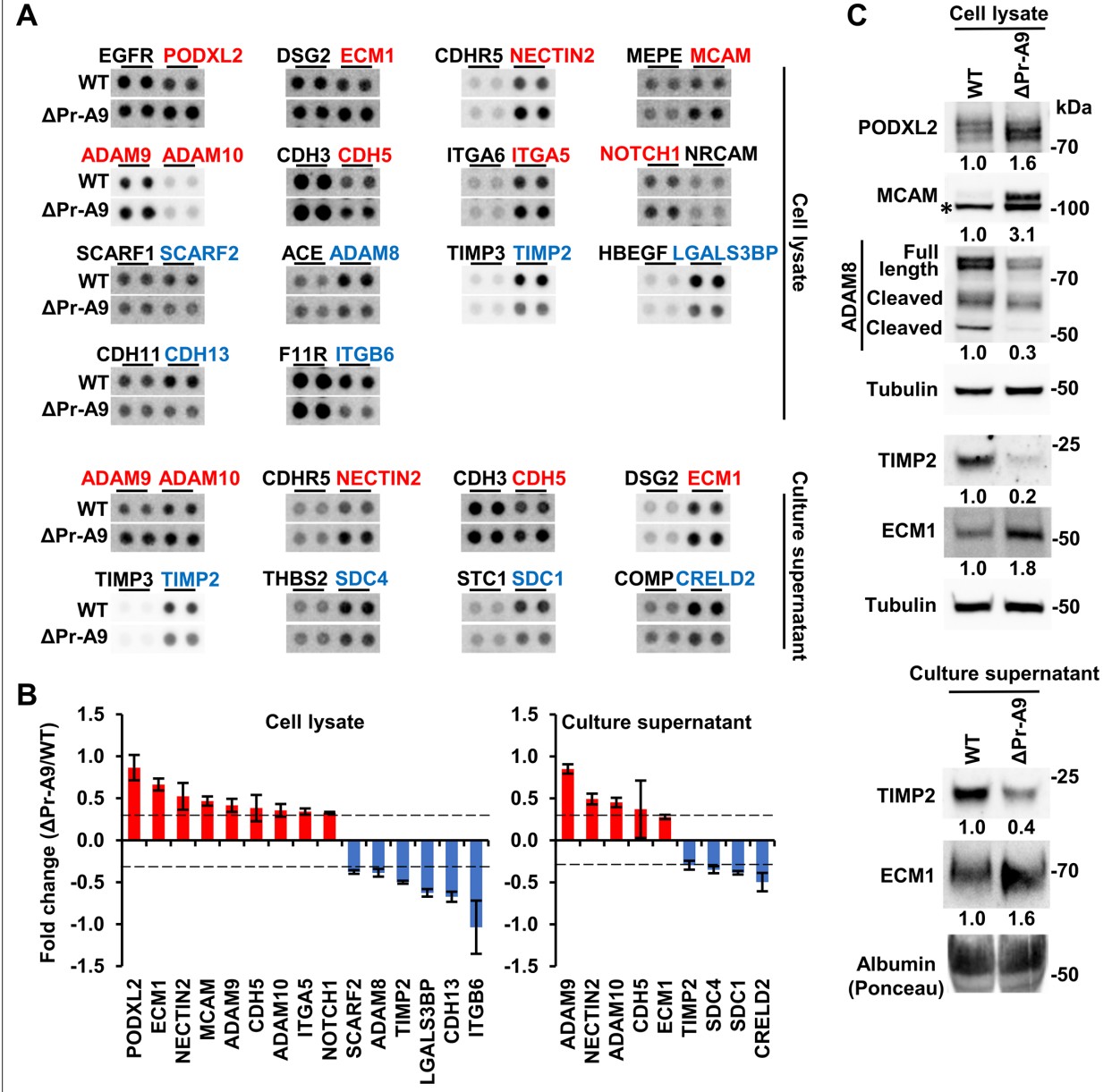

**Figure 4.** Knockout (KO) of lnc-FANCI-2 in CaSki cells affects expression of cellular soluble receptors. (**A**) Dot blots show differentially expressed soluble receptors in ΔPr-A9 cells when compared to those in the wild-type (WT) CaSki cells. A Proteome Profiler Human sReceptor Array was used to examine 105 cellular soluble receptors. Label in red, upregulated receptors; label in blue, downregulated receptors. (**B**) Quantification of differentially expressed and/or released soluble receptors (**A**) in ΔPr-A9 cells when compared to those in the WT CaSki cells. Error bars represent standard deviation of replicates in each proteome array. The dashed line indicates the threshold of fold change (FC) with a score above or below the threshold (-/+0.3 FC) being determined as differentially expressed and/or released receptors. (**C**) Validation of several differentially expressed soluble receptors by immunoblot analysis using specific antibodies. Tubulin served as an internal loading control. *Nonspecific band in MCAM immunoblot.

The online version of this article includes the following source data for figure 4:

**Source data 1.** Knockout (KO) of lnc-FANCI-2 in CaSki cells affects expression of cellular soluble receptors.

**Source data 2.** Knockout (KO) of lnc-FANCI-2 in CaSki cells affects expression of cellular soluble receptors.

and 39 of 130 genes in EMT (*Figure 6B and C*, *Supplementary file 3*) were significantly upregulated, whereas 37 of 139 genes in IFN-γ response and 24 of 71 genes in IFN-α response were significantly downregulated (*Figure 6D and E*, *Supplementary file 3*). Similar enriched gene sets with increased RAS signaling and EMT and decreased IFN-γ and IFN-α responses were observed in ΔYY1-D5 cells (*Figure 6—figure supplement 1*, *Supplementary file 3*).

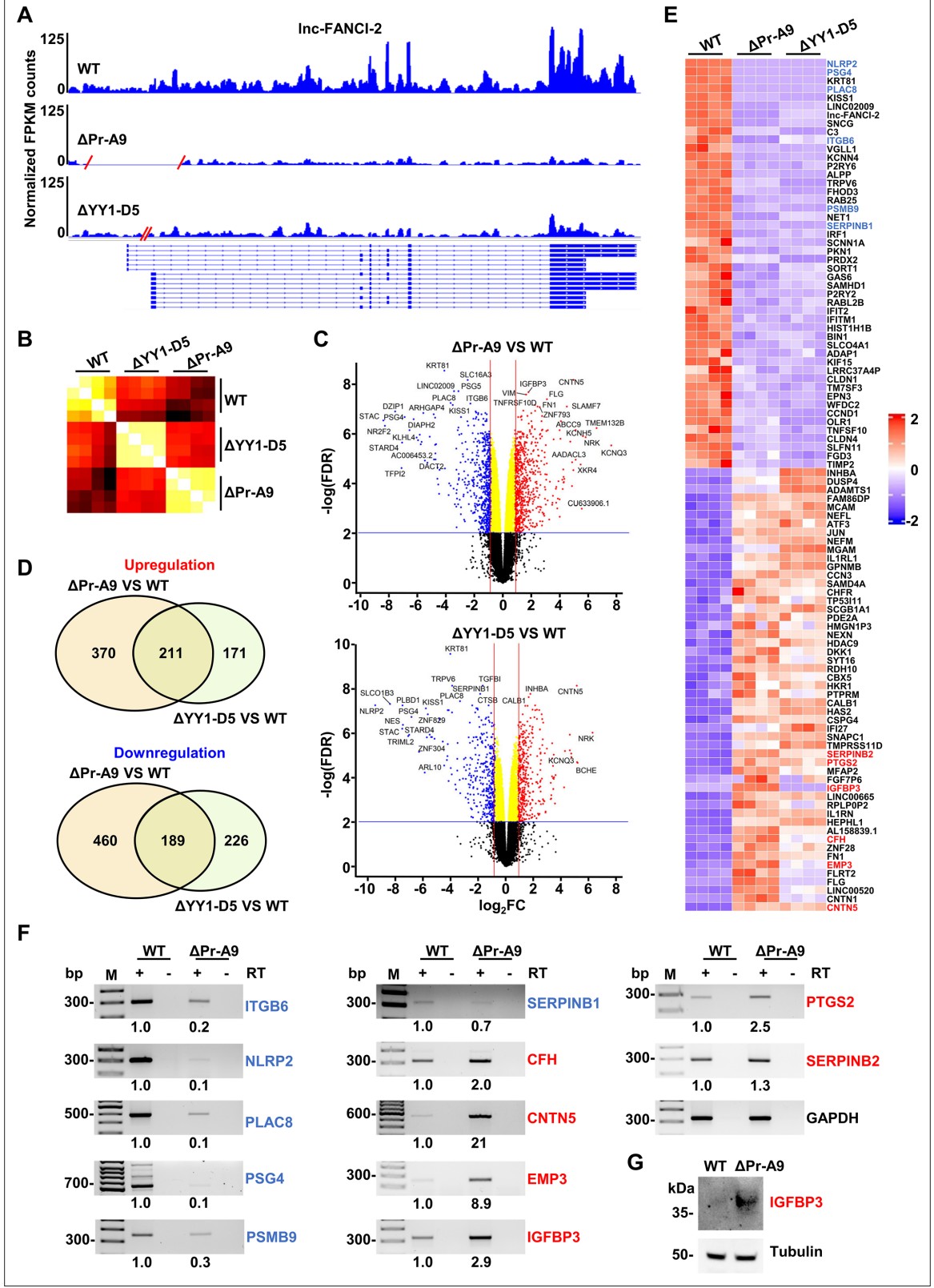

**Figure 5.** Transcriptomic effect of lnc-FANCI-2 knockout (KO) in CaSki cells by RNA-seq analysis. (**A**) RNA-seq reads-coverage maps by Integrative Genomics Viewer (IGV) showing the expression levels of lnc-FANCI-2 from two KO cell clones, ΔPr-A9 and ΔYY1-D5, to the wild-type (WT) CaSki cells. One representative coverage profile of four in each type of cells is shown by IGV. FPKM, fragments per kilobase of transcript per million. Red slashes represent the deleted genomic region, with validated 14 lnc-FANCI-2 RNA isoforms shown below (*Liu et al., 2021*). (**B**) Similarity in heatmap

*Figure 5 continued on next page*

*Figure 5 continued*

comparison among ΔPr-A9, ΔYY1-D5, and WT cells was generated using Limma-normalized counts from each sample by Pearson complete linkage method. (C) Volcano plot visualization of differentially expressed genes (DEGs) in ΔPr-A9 and ΔYY1-D5. The genes with the most significant change in the increased (red) or decreased (blue) expression are indicated. FC, fold change. (D) Venn diagram depicting the overlapped DEGs between ΔPr-A9 and ΔYY1-D5 cells over the WT CaSki cells. (E) Heatmap shows overlapped DEGs with FPKM ≥7 in ΔPr-A9 and ΔYY1-D5 cells when compared with the WT CaSki cells. (F) Validation of selective 12 upregulated or downregulated DEGs shown in the heatmap by RT-PCR with gene-specific primers in the presence (+) or absence (-) of reverse transcriptase (RT). GAPDH served as an internal RNA control. Relative expression level of each gene was calculated based on the band density after normalizing to GAPDH RNA band, with the expression level in the WT CaSki cells set as 1. M, a 100 bp DNA marker. (G) Validation of the upregulated expression of IGFBP3 in ΔPr-A9 cells over the WT CaSki cells by immunoblot analysis.

The online version of this article includes the following source data for figure 5:

**Source data 1.** Validation of selective 12 upregulated or downregulated DEGs by RT-PCR.

**Source data 2.** Validation of selective 12 upregulated or downregulated DEGs by RT-PCR.

Since ΔPr-A9 cells exhibit higher KO efficiency of lnc-FANCI-2 and a higher number of differentially expressed genes (DEGs) and show more severe phenotypic changes in cell proliferation, migration, and colony formation than ΔYY1-D5 cells, the ΔPr-A9 cells were primarily used for further focused studies on RAS signaling in this report.

## lnc-FANCI-2 regulates RAS GTPase activities to phosphorylate RAS signaling effectors Akt and Erk

RAS activation triggers two major downstream signal transduction pathways, Raf/Mek/Erk and PI3K/Akt (*Figure 7—figure supplement 1*), to transduce signals from extracellular stimuli to the cell nucleus where specific effector genes are activated for their corresponding functions (*Conover et al., 2000*; *Allard and Duan, 2018*; *Bach, 2018*; *Downward, 2003*; *Simanshu et al., 2017*). To further confirm the GSEA data and verify that two major signaling pathways of RAS are activated, we first examined and compared the RAS GTPase activities in ΔPr-A9 cells and the WT CaSki cells. We found a significant increase in RAS GTPase activity in ΔPr-A9 cells (*Figure 7A*, top bar graphs), indicating that the endogenous lnc-FANCI-2 RNA in CaSki cells suppresses not only the expression of IGFBP3 (*Figure 5F and G*), the most abundant IGFBP, but also RAS activation (*Figure 6A*, top bar graphs). Since ΔPr-A9 cells had undergone long-term selection and adaption during single-cell screening, the increased RAS GTPase activity might result from selection pressure for cell survival. However, by transient siRNA knockdown (KD) of lnc-FANCI-2 expression in the WT CaSki cells, we obtained a similar, albeit weaker, increase of RAS GTPase activity (*Figure 7A*, lower bar graphs). This result suggests that the increased RAS GTPase activity in ΔPr-A9 cells was unrelated to the persistent selection pressure, but rather the deficiency of lnc-FANCI-2.

Next, we examined activation of the PI3K/Akt and Raf/Mek/Erk pathways (*Castellano and Downward, 2011*; *Asati et al., 2016*; *Cuesta et al., 2021*). We observed a 3-fold increase in phosphorylated Akt (p-Akt) and 2.5-fold increase of p-Erk1/2 (p44/p42 MAPK), after being normalized to total Akt and Erk levels, in ΔPr-A9 cells over the WT CaSki cells (*Figure 7B*). These results were confirmed in ΔYY1-D5 cells (*Figure 7—figure supplement 2A*). The increased p-Akt and p-Erk (mostly p-Erk2/p42) was accompanied by elevated expression of MCAM, VIM, and CCND2, and decreased expression of RAC3 (*Figure 7B*, *Figure 7—figure supplement 2A*). These randomly chosen potential RAS effector proteins facilitate soluble receptor functions, cell proliferation, and EMT. Moreover, transient siRNA KD of lnc-FANCI-2 in the WT CaSki cells also led to the increased levels of p-Akt and p-Erk (*Figure 7C*) and increased expression of MCAM and VIM at 48 hr and 96 hr post-transfection (*Figure 7C*). These observations indicate that a transient loss of lnc-FANCI-2 in CaSki cells is sufficient to trigger RAS signaling. The same siRNA transfection in HeLa cells, an lnc-FANCI-2-negative cell line (*Figure 2A and B*), exhibited no effect on p-Akt or p-Erk1/2 levels (*Figure 7—figure supplement 2B*), but in SiHa cells containing mainly nuclear lnc-FANCI-2 (*Figure 2C and D*), enhanced the expression of p-Akt and p-Erk1/2 (*Figure 7—figure supplement 2B*), verifying the specific effect of lnc-FANCI-2 depletion on RAS signaling pathways independently of predominant cellular distribution of lnc-FANCI-2. The lnc-FANCI-2 KO-mediated increase in p-Akt and p-Erk was sensitive to PI3K inhibitor LY294002 (*Vanhaesebroeck et al., 2021*; *Figure 7D*, lanes 4–6) and MEK1/2 inhibitor U0126 (*Favata et al., 1998*; *Figure 7D*, lanes 7–9), respectively. The inhibitors also decreased the expression of VIM and

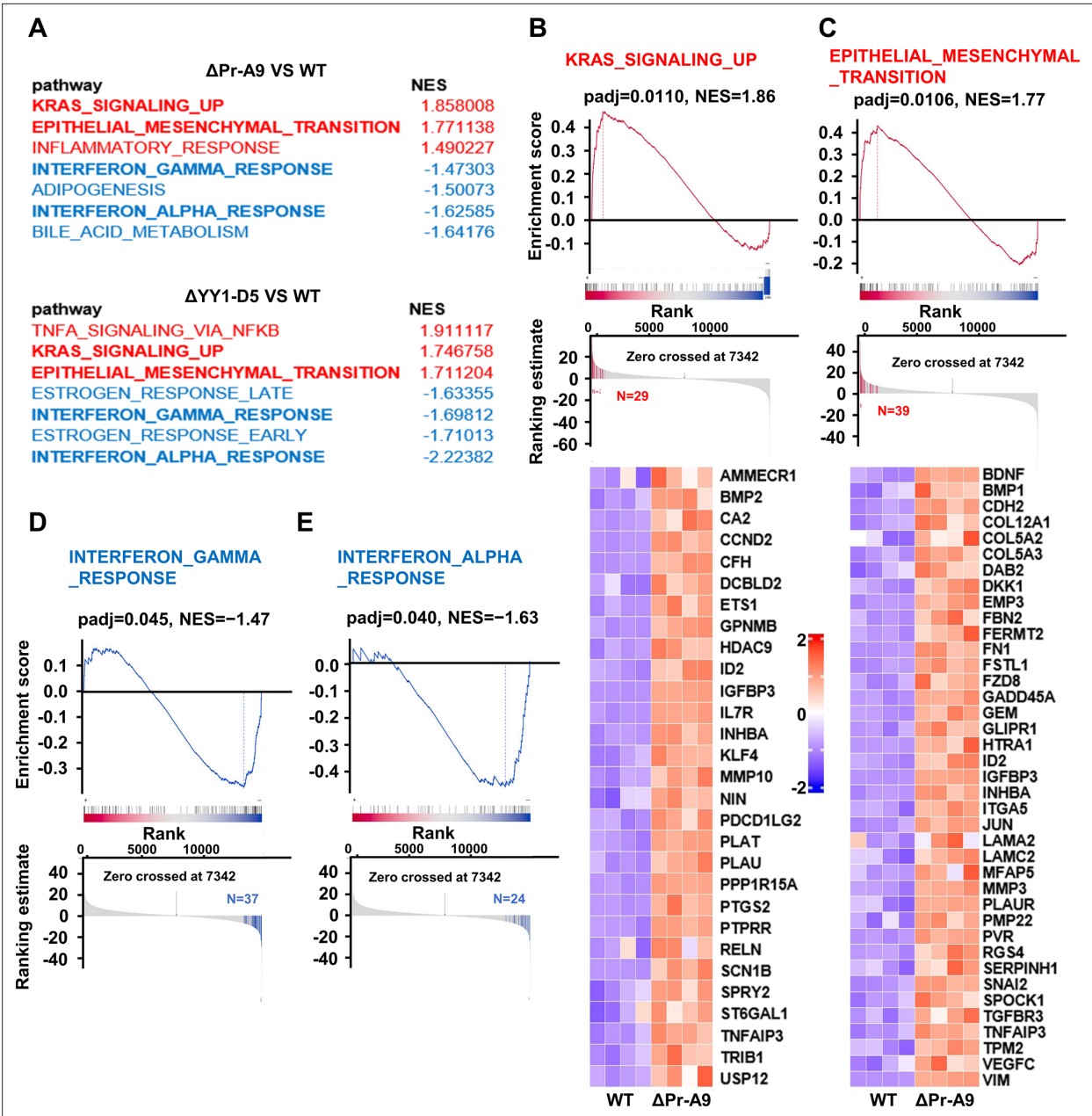

**Figure 6.** Pathway analyses of differentially expressed genes (DEGs) identified by RNA-seq. (**A**) Top 3 upregulated and top 4 downregulated pathways in ΔPr-A9 and ΔYY1-D5 cells over the wild-type (WT) CaSki cells. Data were generated by gene set enrichment analysis (GSEA) performed with Hallmark gene sets. NES stands for normalized enrichment score. (**B**–**E**) Each GSEA plot shows the enrichment score and gene hits enriched in ΔPr-A9 cells using Hallmark gene sets, KRAS_SIGNALING_UP (**B**), EPITHELIAL_MESENCHYMAL_TRANSITION (**C**), INTERFERON_GAMMA_RESPONSE (**D**), or INTERFERON_ALPHA_RESPONSE (**E**). padj, adjusted p-value; Zero crossed 7342, the middle number of total genes in the GSEA at ranking. The heatmaps below the enrichment plots (**B** and **C**) visualize the genes enriched in respective pathways of KRAS_SIGNALING_UP (**B**) and EPITHELIAL_MESENCHYMAL_TRANSITION (**C**) in ΔPr-A9 cells when compared to the WT CaSki cells.

The online version of this article includes the following figure supplement(s) for figure 6:

**Figure supplement 1.** Gene set enrichment analysis (GSEA) enrichment plots show enrichment scores and gene hits enriched in ΔYY1-D5 cells using Hallmark gene sets.

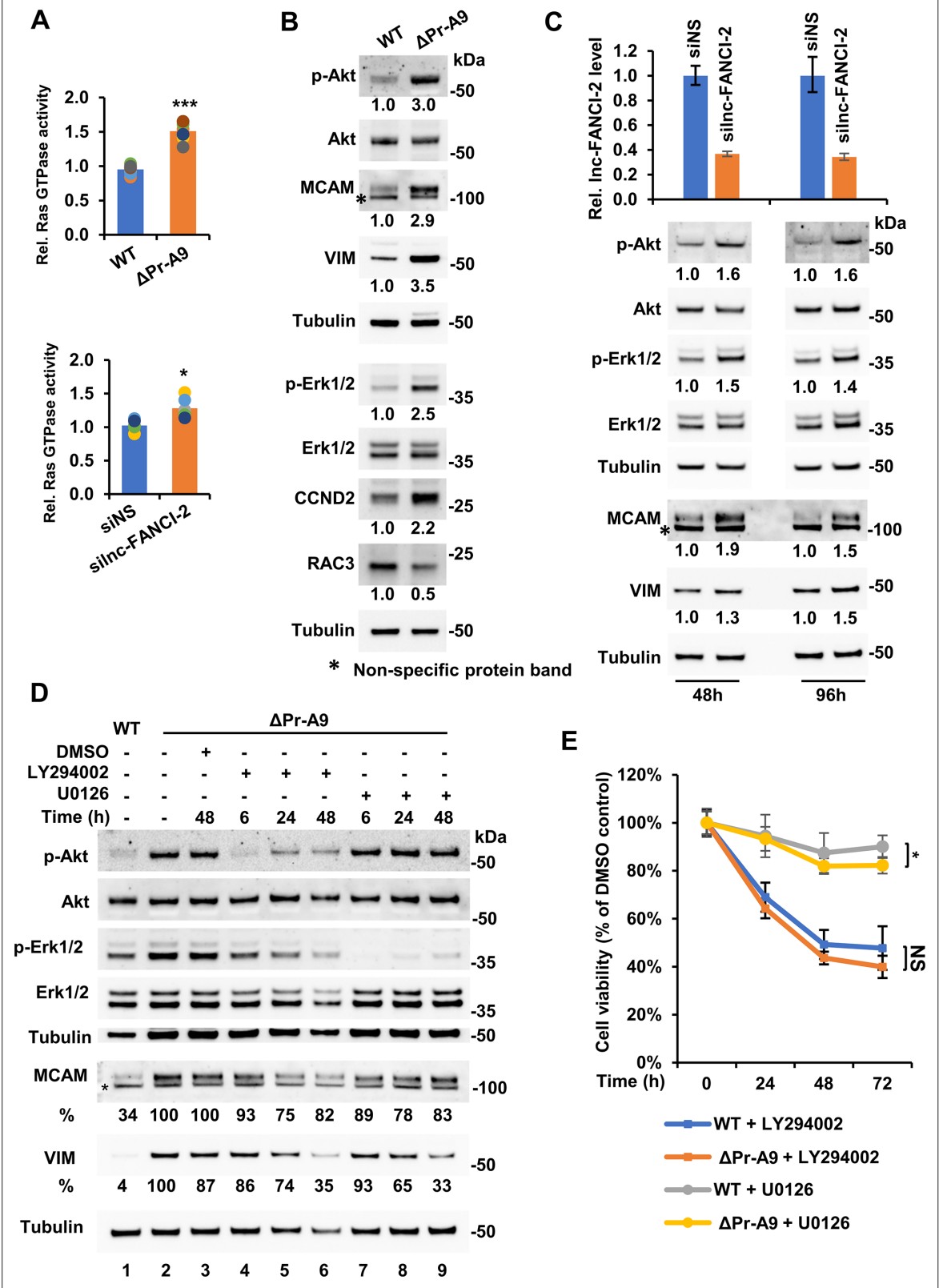

**Figure 7.** CaSki cells with lnc-FANCI-2 knockout (KO) exhibit activation of RAS signaling pathway. (**A**) CaSki cells with lnc-FANCI-2 KO (top, ΔPr-A9 cells) or knockdown (KD, lower) by lnc-FANCI-2-specific siRNAs display increased RAS GTPase activity when compared to the wild-type (WT) CaSki cells (top) or the WT cells treated with a nonspecific control siRNA (siNS). The relative RAS GTPase activity was measured by a RAS GTPase Chemi ELISA. *, p<0.05; ***, p<0.001 by two-tailed Student's t-test. (**B** and **C**) Selective validation of increased expression of RAS signaling-related downstream

*Figure 7 continued on next page*

*Figure 7 continued*

genes in lnc-FANCI-2 KO ΔPr-A9 cells (**B**) over the WT CaSki cells or lnc-FANCI-2 KD CaSki cells (**C**) over the WT cells treated with a nonspecific control siRNA (siNS). Relative expression of the indicated genes in ΔPr-A9 cells in comparison to the WT cells (**B**) or in the WT CaSki cells treated for 48 hr and 96 hr with a nonspecific control siRNA (siNS) or a siRNA specifically targeting the lnc-FANCI-2 exon 3 (**C**) was immunoblotted using the corresponding antibodies as indicated. Tubulin served as an internal control for each blot. The level of p-Akt or p-Erk was calculated by normalizing to total Akt or Erk protein and the other proteins by normalizing to tubulin. The protein level in the WT cells or WT cells treated with siNS is set as 1. The KD efficiency of lnc-FANCI-2 RNA (C, top bar graphs) was examined by RT-qPCR. (**D**) Effect of blocking RAS signaling on the expression of MCAM and VIM using PI3K inhibitor LY294002 (20 μM) and MEK inhibitor U0126 (10 μM). The protein at each time point was immunoblotted with a corresponding antibody. The level of MCAM or VIM was calculated after normalizing to tubulin. The protein level in ΔPr-A9 cells without the inhibitor was set as 100%. *, Nonspecific protein band. (**E**) The time-dependent cell viability of the WT CaSki and ΔPr-A9 cells in the presence of 20 μM LY294002 or 10 μM U0126. Data was obtained at each time point after normalizing to the cells treated with DMSO. The mean ± SD at each data point was calculated from six samples combined from two independent experiments.

The online version of this article includes the following source data and figure supplement(s) for figure 7:

**Source data 1.** lnc-FANCI-2 knockout (KO) and activation of RAS signaling pathway.

**Source data 2.** lnc-FANCI-2 knockout (KO) and activation of RAS signaling pathway.

**Figure supplement 1.** Pathway map of the RAS Initiative with highlighted differentially expressed genes (DEGs) in ΔPr-A9 cells vs parental wild-type cells.

**Figure supplement 2.** Knockout or knockdown (KD) of lnc-FANCI-2 affects RAS signaling.

**Figure supplement 2—source data 1.** lnc-FANCI-2 knockout (KO) and activation of RAS signaling pathway.

**Figure supplement 2—source data 2.** lnc-FANCI-2 knockout (KO) and activation of RAS signaling pathway.

MCAM and cell proliferation by MEK1/2 inhibitor U0126 (*Figure 7D and E*). The effects of LY294002 on cell proliferation were similar from ΔPr-A9 to the WT CaSki cells (*Figure 7E*).

## lnc-FANCI-2 inhibits the expression of IGFBP3 and MCAM (CD146 or MUC18)

Given that IGFBP-3 is the most abundant IGFBP among all six IGFBP members in potentiation of IGF action and PI3K/AKT activities (*Conover et al., 2000*) and a reduced expression of IGFBP-3 mRNA level is associated with progression to cervical cancer (*Serrano et al., 2007*), we further investigated the effect of lnc-FANCI-2 on the expression of IGFBP3 as a lnc-FANCI-2 effector gene. As shown in *Figure 5F and G*, *Figure 8—figure supplement 1A*, the increased RNA and protein expression of IGFBP3 appears in the lnc-FANCI-2 KO ΔPr-A9 cells over the parental WT CaSki cells, indicating a suppressive effect of lnc-FANCI-2 on IGFBP3 expression. This suppressive function of lnc-FANCI-2 was further confirmed in lnc-FANCI-2 rescue experiments in the ΔPr-A9 cells (*Figure 8A and B*). By transient expression of one major isoform of lnc-FANCI-2 RNA (a-PA2, GenBank: MT669800.1) (*Liu et al., 2021*) in the ΔPr-A9 cells, lnc-FANCI-2 (red) and IGFBP3 RNA (green) at 24 hr post transfection were detected by RNAscope RNA-ISH using each specific antisense RNA probe (*Figure 8A*). We demonstrated that the cells with rescued expression of cytoplasmic lnc-FANCI-2 RNA displayed much reduced expression of cytoplasmic IGFBP3 RNA (*Figure 8B*).

As shown in *Figure 5E* and *Figure 8—figure supplement 1B*, the increased RNA expression of MCAM (CD146 or MUC18) (*Wang et al., 2020*) appears to be transcriptional or posttranscriptional in both ΔPr-A9 and ΔYY1-D5 cells. We confirmed by RT-PCR the lnc-FANCI-2 KO-mediated increase of MCAM RNA expression in ΔPr-A9 cells (*Figure 8C*). A rescue experiment was further performed by transient expression of one major isoform of lnc-FANCI-2 RNA (a-PA2, GenBank: MT669800.1) (*Liu et al., 2021*) in ΔPr-A9 cells to determine both nuclear and cytoplasmic MCAM expression (*Kebir et al., 2010*; *Stalin et al., 2016*) in individual lnc-FANCI-2-expressing cells. ΔPr-A9 cells ectopically expressing diffused cytoplasmic lnc-FANCI-2 RNA, as detected by RNAscope RNA-ISH, exhibited a marked reduction in nuclear MCAM protein (*Figure 8D and E*). Quantitative analyses of the ΔPr-A9 cells with or without transient lnc-FANCI-2 RNA expression showed significant reduction of nuclear MCAM protein in the 35 cells with rescued lnc-FANCI-2 expression when compared to the 54 cells expressing no lnc-FANCI-2 (*Figure 8F*). Due to RNAscope procedures, the membrane-bound and cyto-plasmic MCAM protein was mostly removed by protease III treatment when RNA-ISH was performed, and thus only the nuclear signal of MCAM protein that remained was detectable by IF staining. Using cell fractionation, we detected a cleaved MCAM of ~46 kDa mainly in the nuclear fraction by western

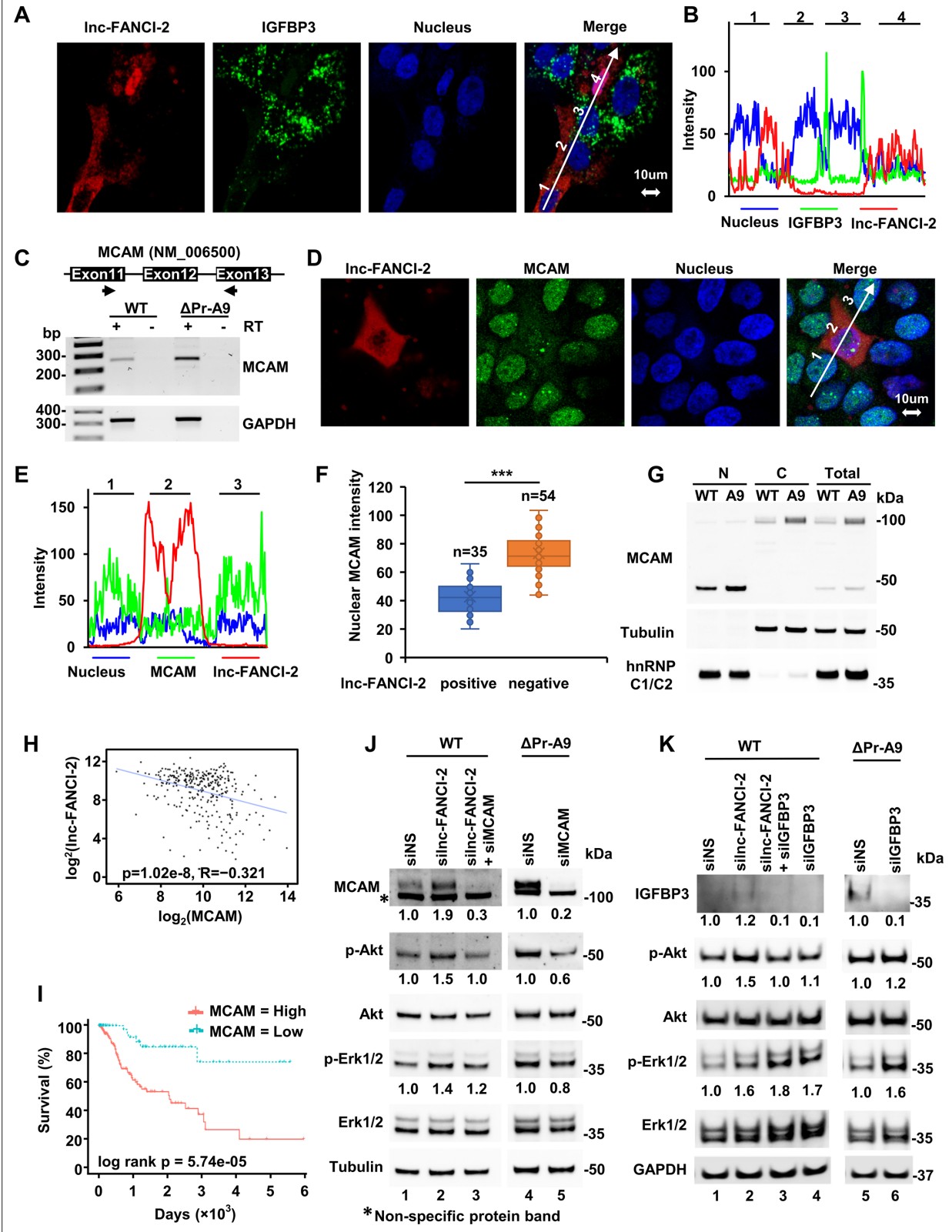

**Figure 8.** lnc-FANCI-2 is suppressive to the expression of IGFBP3 and MCAM. (**A** and **B**) Transient rescue expression of lnc-FANCI-2 in Δpr-A9 cells inhibits the expression of IGFBP3. ΔPr-A9 cells were transfected with a major isoform lnc-FANCI-2a-PA2 (GenBank ACC. No. MT669800.1) cDNA plasmid. lnc-FANCI-2 in red and IGFBP3 RNA in green were detected by RNAscope RNA in situ hybridization (RNA-ISH) at 24 hr post transfection using each specific antisense RNA probe and imaged by confocal microscopy (**A**). Expression levels of lnc-FANCI-2 RNA and IGFBP3 RNA in four neighboring

*Figure 8 continued on next page*

*Figure 8 continued*

cells (**B**) were measured by signal intensity of a line crossing over the stained cells in A (white line arrow). (**C**) Validation of differential expression of MCAM RNA in ΔPr-A9 cells by RT-PCR in the presence (+) or absence (-) of reverse transcriptase (RT). One pair of primers with one primer at exon 11 and the other at exon 13 of MCAM RNA (NM_006500) were used. GAPDH served as a loading control. (**D**) The major isoform lnc-FANCI-2a-PA2 repressed the expression of MCAM. ΔPr-A9 cells were transfected with lnc-FANCI-2a-PA2 cDNA plasmid. lnc-FANCI-2 RNA in red was detected by RNAscope ISH and MCAM protein in green was detected by IF with an anti-MCAM antibody. (**E**) Expression levels of lnc-FANCI-2 RNA and MCAM protein in three neighboring cells were measured by signal intensity of a line crossing over the stained cells (**D**, white line arrow). (**F**) Calculation of the expression levels of MCAM in lnc-FANCI-2-positive cells (n=35) and lnc-FANCI-2-negative cells (n=54) by fluorescent intensity. (**G**) Subcellular MCAM distributions in the nucleus (N) and cytoplasm (C) of WT CaSki cells and ΔPr-A9 by immunoblot analysis. Fractionation efficiency and sample loading were controlled by cytoplasmic (represented by tubulin) and nuclear (represented by hnRNP C1/C2) proteins. (**H–I**) Correlation and survival analysis of lnc-FANCI-2 and MCAM expression with cervical squamous cell carcinoma (CESC) cases from the cancer genome atlas (TCGA) datasets by GEPIA web server (http://gepia.cancer-pku.cn/). The negative correlation at R=–0.321 (with p-value = 1.02e-08) of lnc-FANCI-2 with MCAM in cervical cancer patients was obtained from the RNA-seq data from the TCGA CESC tumors downloaded from the TCGA data portal (https://portal.gdc.cancer.gov/). Only primary solid tumor samples (n=304 after exclusion of 2 metastatic samples and 3 normal samples) were subjected to analysis, with the data showing as a scatter plot (**H**). Kaplan-Meier plot with log-rank test p-value at 5.74e-05 (**I**) shows MCAM as a biomarker for poor prognosis of cervical cancer survival with a lower quartile group cutoff. RNA-seq and survival data are derived from the TCGA CESC cancer patients. (**J/K**) AKT and ERK phosphorylation are partially regulated by MCAM and IGFBP3 in CaSki cells. KD of MCAM (**J**) and IGFBP3 (**K**) expression in WT parental CaSki or ΔPr-A9 cells was treated with MCAM siRNA or IGFBP3 siRNA along with or without lnc-FANCI-2 siRNA for 48 hr. Expression levels of individual proteins were immunoblotted using the corresponding antibodies. The level of MCAM, p-Akt, or p-Erk1/2 was calculated after normalizing to tubulin. The protein level in a non-targeting siRNA (siNS) control was set as 1.

The online version of this article includes the following source data and figure supplement(s) for figure 8:

**Source data 1.** lnc-FANCI-2 is suppressive to the expression of IGFBP3 and MCAM.

**Source data 2.** lnc-FANCI-2 is suppressive to the expression of IGFBP3 and MCAM.

**Figure supplement 1.** Expression of IGFBP3, MCAM, MAP4K4, and lnc-FANCI-2.

**Figure supplement 1—source data 1.** Expression of IGFBP3, MCAM, MAP4K4, and lnc-FANCI-2.

**Figure supplement 1—source data 2.** Expression of IGFBP3, MCAM, MAP4K4, and lnc-FANCI-2.

**Figure supplement 2.** Expression levels of lnc-FANCI-2 and MCAM are significantly associated with the cancer genome atlas (TCGA) cervical squamous cell carcinoma (CESC) cancer patients' survival outcome.

blot (*Figure 8G*). The increased nuclear MCAM expression in ΔPr-A9 cells was proportional to the cytoplasmic and total MCAM levels when compared with the WT CaSki cells (*Figure 8G*). The nuclear MCAM signal appeared as a proteolytic cleavage product presumably by the increased metalloprotease ADAM9 and ADAM10 proteins and decreased metalloprotease inhibitor TIMP2 (*Figure 4*), as observed in other studies (*Stalin et al., 2016*).

The significance of an inverse correlation of lnc-FANCI-2 and MCAM in CaSki cells was further investigated by analyzing their expression in 304 cervical cancer samples from the TCGA dataset. There was a significant negative correlation in lnc-FANCI-2 (being wrongly assigned as LINC00925 by NCBI) (*Liu et al., 2021*) and MCAM RNA levels (*Figure 8H*). The opposite effects of lnc-FANCI-2 and MCAM on cervical cancer survival (*Figure 8I* and *Figure 8H* and *Figure 8—figure supplement 2*) show that the cervical cancer patients with a higher level of lnc-FANCI-2 (*Liu et al., 2021*) but a lower level of MCAM in the cervical tissue (*Figure 8H and I* and *Figure 8—figure supplement 2*) exhibited a better survival prognosis.

## Regulatory roles of lnc-FANCI-2-mediated increase of MCAM and IGFBP3 on RAS signaling

To further dissect the lnc-FANCI-2-associated expression of MCAM and IGFBP3 on RAS signaling, we examined MCAM and IGFBP3 on phosphorylation of Akt and Erk1/Erk2 in CaSki cells in the presence or absence of lnc-FANCI-2. As shown in *Figure 8J*, we found, although KD or KO of lnc-FANCI-2 promotes phosphorylation of Akt and Erk (*Figure 8J*, lane 2), that KD of MCAM expression in the parental WT CaSki cells in the absence of lnc-FANCI-2 (*Figure 8J*, lane 3) or in ΔPr-A9 cells (*Figure 8J*, lane 5) led to reduction of Akt and Erk phosphorylation. These data suggest that MCAM is an lnc-FANCI-2 effector but also could be a trigger of signal transduction as reported (*Wang et al., 2020*; *Joshkon et al., 2020*).

IGFBP3 protein has been viewed as a RAS signaling regulator (*Conover et al., 2000*; *Allard and Duan, 2018*; *Bach, 2018*), but recent studies show that IGFBP3 has a variety of intracellular ligands

involved in many unexpected functions (*Baxter, 2013*; *Varma Shrivastav et al., 2020*). Thus, KD or KO of lnc-FANCI-2-mediated IGFBP3 expression on RAS signaling in the parental WT CaSki cells or ΔPr-A9 cells was examined by western blot after siRNA KD of IGFBP3 expression. As shown in *Figure 8K*, KD of IGFBP3 expression was found to increase phosphorylation of Erk1/2 by ~70% (lane 4) to ~60% (lane 6), but not much so for Akt phosphorylation (lanes 4 and 6). Instead, KD of IGFBP3 expression could prevent lnc-FANCI-2 KD-enhanced Akt phosphorylation in the parental WT CaSki cells (*Figure 8K*, compare lane 2 and lane 3). These data suggest a separate role of IGFBP3 on phosphorylation of Erk1/2 from Akt in the presence or absence of lnc-FANCI-2.

## MAP4K4 association with lnc-FANCI-2 RNA in CaSki cells regulates RAS signaling and phosphorylation of Akt and Erk

We speculated lnc-FANCI-2 restriction on RAS signaling through interactions with cellular proteins. Subsequently, comprehensive identification of lnc-FANCI-2-binding proteins in the parental WT CaSki cells was performed by an isolation of lnc-FANCI-2 RNA-protein complex by RNA purification (IRPCRP) protocol (modified from published ChIRP; *Chu and Chang, 2018*) in combination with mass spectrometry (*Figure 9A*). In this protocol, the lnc-FANCI-2-binding proteins in the parental WT CaSki cells were covalently cross-linked to lnc-FANCI-2 RNA via UV irradiation and pulled down by pooled 30 antisense oligos crossing over the entire lnc-FANCI-2. The RNA extracted from the pull-downs was verified to be lnc-FANCI-2 specific by RT-PCR in the absence (-) or presence (+) of reverse transcriptase (RT) (*Figure 9B*). The lnc-FANCI-2-binding proteins in the pull-downs were then subjected to LC-MS/MS analyses. We identified 32 specific lnc-FANCI-2-binding proteins, including H13, HNRH1, K1H1, MAP4K4, and RNPS1 (*Figure 9C*, *Supplementary file 4*).

MAP4K4, a serine/threonine protein kinase, stimulates cancer cell proliferation, invasion, and migration and is recently characterized as a novel MAPK/ERK pathway regulator (*Gao et al., 2016*; *Gao et al., 2017*; *González-Montero et al., 2023*) and negatively regulates RAS signaling by binding to Ras p21 protein activator 1 or RASA1 (*Patterson et al., 2023*; *Roth Flach et al., 2016*). Although the exact MAP4K4 binding site on lnc-FANCI-2 was not explored, many enzymes turn out to be an RNA-binding enzyme (*Castello et al., 2015*; *Hentze et al., 2018*). Thus, whether MAP4K4 association with lnc-FANCI-2 could regulate PI3K/Akt or MAPK/Erk signal transduction was investigated in the parental WT CaSki cells. We demonstrated that siRNA KD of MAP4K4 expression in the parental WT CaSki cells, as seen for KD of lnc-FANCI-2, led to ~40% or more increase of phosphorylation of both Akt and Erk1/2 and increased expression of MCAM and IGFBP3 (*Figure 9D*, compare lane 1 to lanes 2 and 3). Interestingly, siRNA KD of MAP4K4 expression in the lnc-FANCI-2 KO ΔPr-A9 cells had no effect on expression of IGFBP3, only a minimal effect on MCAM and p-Akt, but strongly on pErk1/2 (*Figure 8—figure supplement 1C*). The data suggest that, through RNA-protein interactions, the increased lnc-FANCI-2 RNA in cells and their association with MAP4K4 and other cellular proteins could be one arm to control RAS signaling and gene expression of its effectors (*Figure 9E*).

## Discussion

HPV oncoproteins E6 and E7 are necessary but not sufficient for development of cervical cancer. Indeed, early studies indicated that other oncogenic stress, such as activated RAS mutant, is needed to trigger malignant transformation of E6- and E7-expressing cells (*Storey et al., 1988*; *Phelps et al., 1988*; *Crook et al., 1988*; *Matlashewski et al., 1987*) and tumorigenesis (*DiPaolo et al., 1989*; *Schreiber et al., 2004*; *Greenhalgh et al., 1994*; *Riou et al., 1988*; *Eiben et al., 2002*; *Golijow et al., 1999*; *Prokopakis et al., 2002*). Moreover, E7 has also been reported to repress phosphorylation of Akt and Akt-mediated signaling (*Strickland and Vande Pol, 2016*). In this study, we show that lnc-FANCI-2, whose expression is highly dependent on E7 and YY1 (*Liu et al., 2021*), intrinsically restricts RAS GTPase activities and phosphorylation of Akt and Erk presumably by interacting with cellular factors in HPV16-positive CaSki cells. Interestingly, early reports indicated that p-Akt attenuation by E7 could be abolished by introduction of the H73E mutation in the E7 CR3 domain (*Strickland and Vande Pol, 2016*). This E7 CR3 domain is also essential for interacting with transcription factor YY1 to activate lnc-FANCI-2 transcription (*Liu et al., 2021*). Thus, it would be presumable that the reported p-Akt attenuation by E7 might be mediated by the increased expression of lnc-FANCI-2. More importantly, these findings also suggest that additional oncogenic stress, such as activated RAS mutant, is

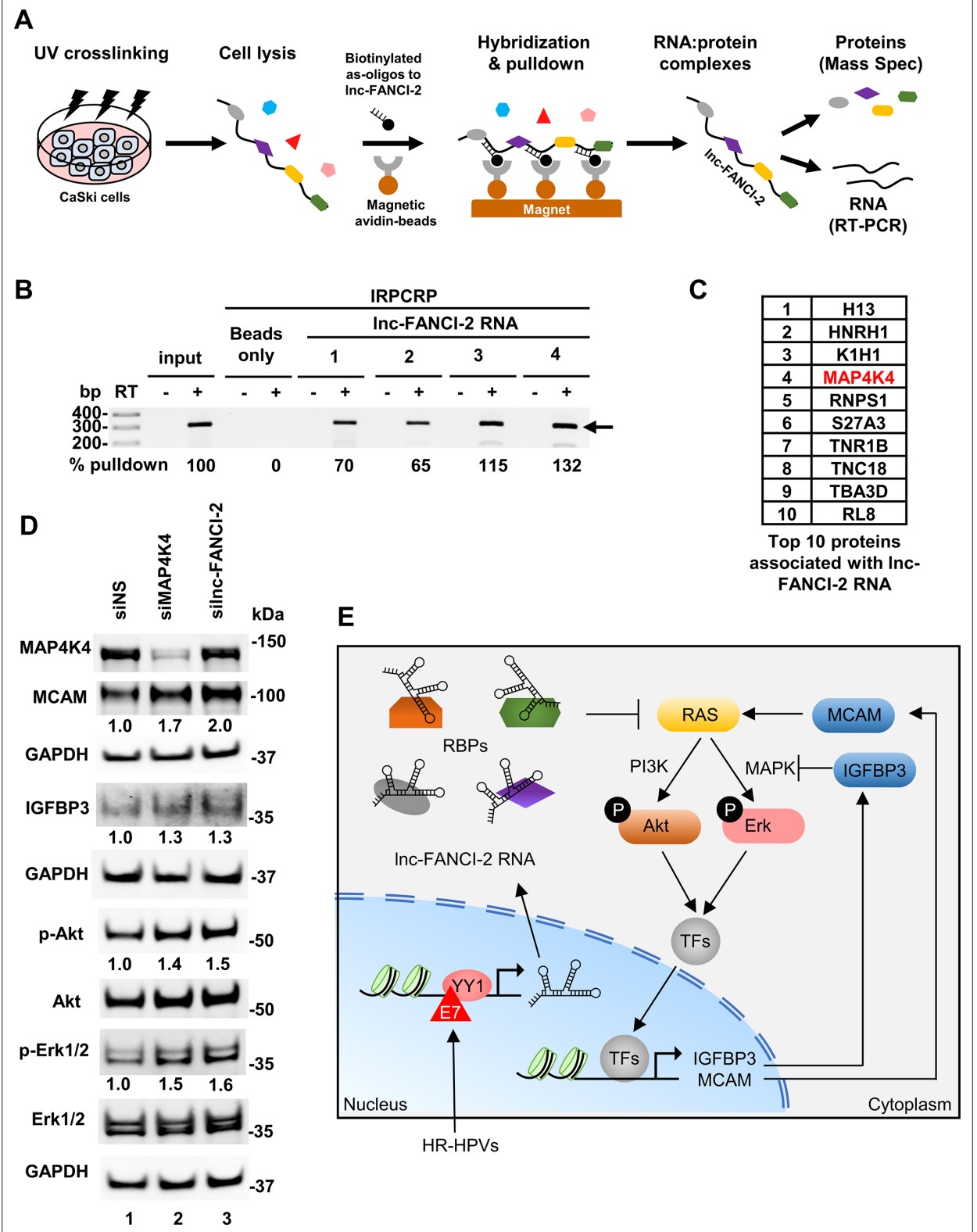

**Figure 9.** lnc-FANCI-2 interacts with host factors to regulate RAS signaling pathway. (**A**) The lnc-FANCI-2-associated proteins in the wild-type (WT) CaSki cells were identified by isolation of RNA-protein complexes using RNA purification (IRPCRP)-mass spectrometry technology. (**B**) lnc-FANCI-2 RNA in the IRPCRP-1 and IRPCRP-2 pull-downs had the pooled antisense biotinylated oligos (pool 1 with oligos in even numbers and pool 2 with oligos in odd numbers) immobilized to avidin-beads first before mixed with cell lysates, while the IRPCRP-3 and IRPCRP-4 pull-downs had the oligos pool 1 and 2 separately mixed with cell lysates first before addition to the avidin-beads for the RNA pull-downs. RT-PCR in the absence (-) or presence (+) of

*Figure 9 continued on next page*

*Figure 9 continued*

reverse transcriptase (RT) was carried out using the RNA isolated from the individual IRPCRP experiments using a primer pair of oHBL5 and oHBL12 (*Supplementary file 5*) specific for lnc-FANCI-2 RNA detection. Beads-only (no oligos) IRPCRP experiments served as a negative control. Total RNA from the WT CaSki cells after sonication was used as an input control. The arrow indicates the detected lnc-FANCI-2 RNA. (**C**) Proteins associated with lnc-FANCI-2 RNA identified from lnc-FANCI-2 IRPCRP pull-downs. A total of 32 proteins were specifically pulled down from lnc-FANCI-2 IRPCRP reactions 1–4 (PSM ≥2 from two separate pull-downs, *Supplementary file 4*), with top 10 proteins binding lnc-FANCI-2 shown in the order by the number of identified peptide spectrum match (PSM) in LC-MS/MS. (**D**) Expression of p-Akt and p-Erk from CaSki WT cells 48 hr after siRNA KD of MAP4K4 or lnc-FANCI-2 was immunoblotted by the corresponding antibodies. GAPDH served as a sample loading control. Fold change of the indicated proteins in the cells with KD of MAP4K4 or lnc-FANCI-2 over the cells treated by a non-targeting siRNA (siNS) was calculated after normalizing to GAPDH. (**E**) A proposed model illustrates how lnc-FANCI-2-protein complexes inhibit the RAS signaling pathway to control Akt/Erk phosphorylation and expression of host genes. RAS signaling can be regulated by integrating external and internal factors. In high-risk human papillomavirus (HR-HPV) infected cells, viral oncoprotein E7-YY1 complex transactivates lnc-FANCI-2 expression. By interactions with cellular RBPs, the lnc-FANCI-2-protein complex inhibits RAS signaling. In the absence of lnc-FANCI-2, increased RAS signaling leads to phosphorylation of Akt and Erk and cascaded responses of transcription factors (TFs) and thus regulates the expression of RAS signaling responder genes, such as IGFBP3, MCAM, etc. Consequently, this brings fundamental biochemical and biological processes under control by fine-tuning the RAS pathway.

The online version of this article includes the following source data for figure 9:

**Source data 1.** lnc-FANCI-2 interacts with host factors to regulate RAS signaling pathway.

**Source data 2.** lnc-FANCI-2 interacts with host factors to regulate RAS signaling pathway.

required to overcome the lnc-FANCI-2 restriction on RAS signaling for the early observed malignant transformation (*Storey et al., 1988*; *Phelps et al., 1988*; *Crook et al., 1988*; *Matlashewski et al., 1987*) and tumorigenesis (*DiPaolo et al., 1989*; *Schreiber et al., 2004*; *Greenhalgh et al., 1994*; *Riou et al., 1988*; *Eiben et al., 2002*; *Golijow et al., 1999*; *Prokopakis et al., 2002*) of E6- and E7-expressing cells. In CaSki cells, loss of lnc-FANCI-2 in this report was found to promote RAS signaling but reduction of IFN responses. However, high RAS-AKT-ERK signaling induces cell senescence and inhibits tumor cell survival as shown in this report and in a lung cancer study (*Nieto et al., 2017*).

Some lncRNAs function as RAS regulators (*Yang et al., 2019*; *Saliani et al., 2022*) by their ability to sequester RAS-targeting miRNAs, including MALAT1/miR-217, RMRP/miR-206, and KRAS1P/miR-143/let-7 (*Saliani et al., 2022*). Conceivably, lncRNAs may also regulate RAS signaling through other mechanisms beyond miRNAs. In this report, loss of lnc-FANCI-2 in HPV16-positive CaSki and SiHa cells promotes RAS signaling and expression of RAS signaling effectors (*Figure 9E*). By profiling the proteins associated with lnc-FANCI-2 in CaSki cells, we identified a serine/threonine protein kinase MAP4K4 as one of the major lnc-FANCI-2-binding proteins that may partially mediate the lnc-FANCI-2 restriction on RAS signaling. MAP4K4 is a negative RAS signaling regulator in the context of the early embryo (*Gao et al., 2017*; *Patterson et al., 2023*) and lymphatic vascular development by interacting with RASA1 (*Roth Flach et al., 2016*). Indeed, silencing of MAP4K4 in parental WT CaSki cells, like KD or KO of lnc-FANCI-2, led to a partial increase of both p-Akt and p-Erk1/2 (*Figure 9D*) as reported (*Patterson et al., 2023*; *Roth Flach et al., 2016*). Interestingly, KD of MAP4K4 in ΔPr-A9 cells led to further increased expression of p-Erk1/2 but minimal on p-Akt (*Figure 8—figure supplement 1C*), suggesting a negative correlation of MAP4K4 activity in interaction with lnc-FANCI-2 on Erk1/2 phosphorylation. Thus, our data suggest that MAP4K4 protein may function in coordination with lnc-FANCI-2 RNA in CaSki cells to restrict one arm of RAS signaling, but not the other.

All HR-HPVs are capable of inducing lnc-FANCI-2 expression by experimental infection of keratinocytes (*Liu et al., 2021*). HPV+ cervical cancer tissues express high levels of lnc-FANCI-2. As expected, all HPV16+ cervical cancer and pre-cancer cell lines examined – such as CaSki, SiHa, W12 subclone cells 20861 (HPV16+, integrated) and 20863 (HPV16+, episomal) – produce lnc-FANCI-2. Unexpectedly, we found no expression of lnc-FANCI-2 in isogenic HPV18+ HeLa and C4II cells, nor in colorectal cancer cell line HCT116, adenovirus 5 DNA-transformed human embryonic kidney cell line HEK293, spontaneously transformed aneuploid immortal keratinocyte cell line HaCaT, and KSHV-infected B cell lymphoma cell line BCBL-1 cells which are all HPV-negative cell lines. It remains to know what negatively regulates lnc-FANCI-2 expression in those cell lines, and more HPV18+ cell lines should be examined for lnc-FANCI-2 expression. However, we did find that an HPV-negative cervical cancer cell line, C33A cells expressing mutant p53 and mutant pRB (*Scheffner et al., 1991*), produces a high level of lnc-FANCI-2 (*Figure 2*). Using dual luciferase report assays, we did find lnc-FANCI-2 promoter activity in response to YY1 binding in CaSki and C33A cells, but not in HeLa and HCT116 cells, suggesting

the presence of a repressive factor(s) in the latter two cell lines (data not shown). KD of E2F1 had no effect on lnc-FANCI-2 promoter activity in CaSki cells (*Liu et al., 2021*). In the lnc-FANCI-2-producing cells, we noticed that the cellular location of lnc-FANCI-2 also varies from cell types, with lnc-FANCI-2 predominantly in the nucleus in SiHa and C33A cells, but in the cytoplasm in CaSki cells, although both cytoplasmic and nuclear lnc-FANCI-2 restrict RAS signaling. This suggests that a variety of nuclear and cytoplasmic functions of lnc-FANCI-2 remains to be explored.

IGFBP3 is a major IGF carrier and enhances both IGF- and EGF-RAS signaling (*Allard and Duan, 2018*; *Martin et al., 2003*) to activate phosphatidylinositol 3-kinase (PI3K)/AKT and the MAPK/ERK pathways (*Józefiak et al., 2021*). Surprisingly, we find a suppressive role of IGFBP3 on Erk phosphorylation, but not much on Akt, in the presence or absence of lnc-FANCI-2 in HPV16-positive CaSki cells. Although ~99% of circulating IGF-1 is bound to IGFBPs, predominantly to IGFBP-3, the most abundant IGFBP in human serum, case-control studies showed significantly lower IGF-1 and IGFBP-3 serum levels in the patients with ICC over the control group (*Jozefiak et al., 2008*; *Serrano et al., 2006*). A reduced expression of IGFBP-3 mRNA level appears to be associated with progression to cervical cancer (*Serrano et al., 2007*). Consistent with lnc-FANCI-2's suppressive effect on IGFBP3 expression in our study, an increased level of lnc-FANCI-2 expression is associated step-wisely with HR-HPV-positive CIN and ICC tissues (*Liu et al., 2021*).

It is obvious from our RNA-seq and human soluble receptor array analysis that KO of lnc-FANCI-2 in CaSki cells affects the expression of a large set of genes, including the expression of increased NECTIN2, ADAM9, ADAM10, ITGA5, MCAM, PODXL2, ECM1, but decreased ADAM8, TIM2, and ITGB6, which are responsible for RAS signaling, EMT pathway, and IFN responses. For example, the dysregulation of RAS signaling and ADAM protein activity is implicated in various cancers. ADAM proteins can modulate RAS signaling by cleaving and releasing ligands that activate or inactivate RAS-related pathways (*Schäfer et al., 2004*; *Ohtsu et al., 2006*; *Dang et al., 2011*; *Kleino et al., 2015*). Some ADAM proteins are involved in the migration and invasion of cancer cells, and their loss can promote the degradation of KRAS (*Huang et al., 2024*). ADAMs are essential for extracellular signaling and regulation of cell adhesion, and their catalytic activity is strongly correlated to Src-homology 3 (SH3) domain binding of SH3 proteins (*Dang et al., 2011*; *Kleino et al., 2015*; *Sun et al., 2010*). In addition to the notified activation of PI3K/Akt and Raf/Mek/Erk pathways, one of the major RAS signaling effectors in lnc-FANCI-2 KO or KD cells was found to be MCAM.

MCAM binds various cellular surface receptors or co-receptors to trigger signal transduction, cell proliferation and motility, tumor angiogenesis, and tumor cell metastasis (*Wang et al., 2020*; *Joshkon et al., 2020*). It is an integral, highly glycosylated membrane protein with three common protein isoforms (*Wang et al., 2020*; *Anfosso et al., 1998*; *Dye et al., 2009*): two membrane-anchored forms with a long (p113) or short (p76) cytoplasmic tail produced by alternative RNA splicing and a proteolytically cleaved soluble form without transmembrane and cytoplasmic regions (*Wang et al., 2020*; *Dye et al., 2009*; *Bardin et al., 1998*). In addition to its roles in cell signal transduction (*Wang et al., 2020*; *Joshkon et al., 2020*; *Anfosso et al., 1998*; *Dye et al., 2009*), the short isoform produced by an intramembrane cleavage may be directed toward the nucleus to regulate gene transcription (*Stalin et al., 2016*). We found that CaSki cells express mainly a cytoplasmic form of full-length MCAM (~113 kDa) and a small nuclear form (~46 kDa), but not the short (~76 kDa) isoform. Moreover, lnc-FANCI-2 KO cells exhibit high MCAM levels, which was most likely a result of the increased RAS signaling. Consistently, lnc-FANCI-2 and MCAM levels are negatively correlated in cervical cancer patients, and low levels of MCAM and high levels of lnc-FANCI-2 indicate better prognosis. This suggests that increased MCAM might promote cervical tumorigenesis in vivo. Given that the cells with lnc-FANCI-2 KO exhibit increased levels of IGFBP3 and MCAM, and the ΔPr-A9 cells with rescued lnc-FANCI-2 RNA expression displayed remarkable reduction of IGFBP3 and MCAM expression, our current model (*Figure 9E*) highlights how the HPV16-enhanced expression of lnc-FANCI-2 in cervical cancer cells might function as a negative regulator by binding to various RNA-binding proteins to block RAS signaling and reduce the expression of RAS signaling effectors such as IGFBP3 and MCAM.

In conclusion, this study provides the first evidence that one of lnc-FANCI-2 functions is to maintain epithelial functional integrity by repressing RAS signaling in preventing malignant transformation of neoplastic cervical lesions. In addition, more extensive studies are underway to elucidate the role of lnc-FANCI-2 in regulation of IFN responses, which are not covered in this report. Since transactivation of lnc-FANCI-2 expression is largely dependent on host transcription factor YY1, of which expression

is regulated by HPV oncoproteins and host miR-29a, dissecting their transcription network on lnc-FANCI-2 and its effectors may further provide novel insights involved in host homeostasis against HPV infection-induced carcinogenesis. Our observations also highlight additional, unexpected players beyond viral E6 and E7 that drive cells toward malignant transformation.

# Materials and methods

## Key resources table

| Reagent type (species) or resource | Designation | Source or reference | Identifiers | Additional information |
|---|---|---|---|---|
| Cell line (*Homo sapiens*) | CaSki (HPV16+) | ATCC | CRL-1550 | Cervical cancer cell line |
| Cell line (*Homo sapiens*) | C4II (HPV18+) | ATCC | CRL-1595 | Cervical cancer cell line |
| Cell line (*Homo sapiens*) | SiHa (HPV16+) | ATCC | HTB-35 | Cervical cancer cell line |
| Cell line (*Homo sapiens*) | HeLa (HPV18+) | ATCC | CCL-2 | Cervical cancer cell line |
| Cell line (*Homo sapiens*) | C33A (HPV) | ATCC | HTB-31 | Cervical cancer cell line |
| Cell line (*Homo sapiens*) | HEK293 | ATCC | CRL-1573 | Kidney epithelial-like cell |
| Cell line (*Homo sapiens*) | HCT116 | Dr. Bert Vogelstein (Johns Hopkins University) | N/A | Colorectal cancer cell line |
| Cell line (*Homo sapiens*) | W12 subclones (20861, 20863) | Dr. Paul Lambert (University of Wisconsin) | N/A | HPV16+ (integrated/episomal) |
| Cell line (*Homo sapiens*) | BCBL-1 (KSHV+) | NIH AIDS Reagent Program | N/A | Body cavity B lymphoma |
| Cell line (*Homo sapiens*) | HaCaT | Dr. Carl C Baker (NIH) | N/A | Skin keratinocyte |
| Antibody | Anti-MCAM (Rabbit polyclonal) | ProteinTech | Cat# 17564-1-AP | Rabbit polyclonal; WB (1:1000), IF (1:200) |
| Antibody | Anti-E2F1 (Rabbit polyclonal) | ProteinTech | Cat# 12171-1-AP | Rabbit polyclonal; WB (1:1000) |
| Antibody | Anti-MAP4K4 (Rabbit polyclonal) | ProteinTech | Cat# 55247-1-AP | Rabbit polyclonal; WB (1:1000) |
| Antibody | Anti-p53 (Rabbit polyclonal) | ProteinTech | Cat# 10442-1-AP | Rabbit polyclonal; WB (1:1000) |
| Antibody | Anti-α-Tubulin (Mouse monoclonal) | Sigma-Aldrich | Cat# T5201 | Mouse monoclonal; WB (1:5000) |
| Antibody | Anti-hnRNP C1/C2 (Mouse monoclonal) | Abcam | Cat# Ab10294 | Mouse monoclonal; WB (1:1000) |
| Antibody | Anti-RAC3 (Rabbit monoclonal) | Abcam | Cat# ab129062 | Rabbit monoclonal; WB (1:1000) |
| Antibody | Anti-CCND2 (Cyclin D2) (Rabbit monoclonal) | Cell Signaling Technology | Cat# 3741 | Rabbit monoclonal; WB (1:1000) |
| Antibody | Anti-Akt (Rabbit polyclonal) | Cell Signaling Technology | Cat# 9272 | Rabbit monoclonal; WB (1:1000) |
| Antibody | Anti-phospho-Akt (Ser473) (Rabbit polyclonal) | Cell Signaling Technology | Cat# 9271 | Rabbit monoclonal; WB (1:1000) |
| Antibody | Anti-Erk1/2 (p44/42 MAPK) (Rabbit monoclonal) | Cell Signaling Technology | Cat# 4695 | Rabbit monoclonal; WB (1:1000) |
| Antibody | Anti-phospho-Erk1/2 (Thr202/Tyr204) (Rabbit monoclonal) | Cell Signaling Technology | Cat# 4370 | Rabbit monoclonal; WB (1:1000) |
| Antibody | Anti-IGFBP3 (D1U9C) (Rabbit monoclonal) | Cell Signaling Technology | Cat# 25864 | Rabbit monoclonal; WB (1:1000) |
| Antibody | Anti-Vimentin (VIM) (Mouse monoclonal) | Thermo Fisher Scientific | Cat# MA5-11883 | Mouse monoclonal; WB (1:1000), IF (1:200) |
| Antibody | Anti-PODXL2 (Goat polyclonal) | R&D Systems | Cat# AF1524 | Goat polyclonal; WB (1:500) |
| Antibody | Anti-ECM1 (Mouse monoclonal) | R&D Systems | Cat# MAB39371 | Mouse monoclonal; WB (1:500) |
| Antibody | Anti-TIMP2 (Goat polyclonal) | R&D Systems | Cat# AF971 | Goat polyclonal; WB (1:500) |
| Antibody | Anti-ADAM8 (Goat polyclonal) | R&D Systems | Cat# AF1031 | Goat polyclonal; WB (1:500) |
| Antibody | Anti-SRSF3 (Rabbit monoclonal) | Novus Biologicals | Cat# NBP2-76892 | Rabbit monoclonal; WB (1:1000) |
| Antibody | Anti-HPV16 E7 (Rabbit polyclonal) | GeneTex | Cat# GTX133411 | Rabbit polyclonal; IF (1:500) |

*Continued on next page*

*Continued*

| Reagent type (species) or resource | Designation | Source or reference | Identifiers | Additional information |
|---|---|---|---|---|
| Sequence-based reagent | lnc-FANCI-2 siRNA | Custom (see *Supplementary file 5*) | N/A | Double-stranded, target-specific siRNA |
| Sequence-based reagent | MAP4K4 siRNA | Dharmacon, Inc | #M-003971 | Human MAP4K4 targeting siRNA |
| Sequence-based reagent | IGFBP3 siRNA | Dharmacon, Inc | #M-004777 | Human IGFBP3 targeting siRNA |
| Sequence-based reagent | Non-targeting control siRNA | Dharmacon, Inc | #D-001210-01 | Negative control siRNA |
| Chemical compound, drug | LY294002 (PI3K inhibitor) | Cell Signaling Technology | #9901 | 20 µM treatment |
| Chemical compound, drug | U0126 (MEK inhibitor) | Cell Signaling Technology | #9903 | 10 µM treatment |
| Commercial assay, kit | RAS GTPase Chemi ELISA Kit | Active Motif | #52097 | RAS activity measurement |
| Sequence-based reagent | lnc-FANCI-2 siRNA | Dharmacon | Custom (*Supplementary file 5*) | Target: lnc-FANCI-2 |
| Sequence-based reagent | lnc-FANCI-2 gRNAs | Zhang Lab CRISPR Designer | Custom (*Supplementary file 5*) | CRISPR knockout |
| Software, algorithm | CCBR Pipeliner | GitHub | CCBR/Pipeliner | RNA-seq analysis |
| Software, algorithm | ZEN 2.3 | Zeiss | N/A | Confocal microscopy |
| Commercial assay, kit | Nuclei EZ Prep Kit | Sigma-Aldrich | #NUC-101 | Nuclear/cytoplasmic fractionation |
| Commercial assay, kit | Superscript First-Strand Synthesis kit | Thermo Fisher Scientific | #11904018 | cDNA synthesis |
| Commercial assay, kit | TaqMan gene expression Master Mix | Applied Biosystems | #4369016 | qPCR amplification |
| Commercial assay, kit | TaqMan Gene Expression Assay (GAPDH) | Applied Biosystems | Hs02758991_g1 | Internal control for qPCR |
| Commercial assay, kit | RNAscope Multiplex Fluorescent V2 Assay | Advanced Cell Diagnostics | #323100 | RNA in situ hybridization |
| Sequence-based reagent | lnc-FANCI-2 custom probe | Advanced Cell Diagnostics | Custom (nt 359–1713 of MT669800.1) | Targets lnc-FANCI-2a-PA2 transcript |
| Sequence-based reagent | IGFBP3 probe | Advanced Cell Diagnostics | #310351 | Channel 1 probe |
| Sequence-based reagent | lnc-FANCI-2 probe | Advanced Cell Diagnostics | #509061-C3 | Channel 3 probe |

## Cell cultures and treatment

CaSki, a HPV16-positive (HPV16+) human cervical cancer cell line with WT p53, pRb, HRAS, KRAS, and NRAS (https://depmap.org/portal/cell_line/ACH-001336?tab=mutation), was obtained from the American Type Culture Collection (ATCC, Manassas, VA, USA), maintained in Dulbecco's modified Eagle medium (DMEM) (Thermo Fisher Scientific) supplemented with 10% fetal bovine serum (FBS) and 1% penicillin-streptomycin, and grown in a cell culture incubator at 37°C with 5% $CO_2$.

SiHa (HPV16+), HeLa (HPV18+), C4II (HPV18+), C33A (no HPV), and HEK293 (Ad5 E1/E2-immortalized human kidney cells) were all obtained from ATCC and grown in DMEM with 10% FBS at 37°C and 5% $CO_2$. HaCaT (spontaneously immortalized human epidermal cells) was a gift from Dr. Carl C Baker in NIH cultured in DMEM with 10% FBS at 37°C and 5% $CO_2$. HCT116 cell line, a colorectal cancer cell line, was a gift from Dr. Bert Vogelstein of Johns Hopkins University and was grown in McCoy's 5A medium with 10% FBS at 37°C and in 5% $CO_2$. W12 subclones 20861 (HPV16+, integrated) and 20863 (HPV16+, episomal) were a gift from Dr. Paul Lambert of University of Wisconsin, Madison, and grown on mitomycin C-pretreated NIH3T3 feeder cells in F12 medium at 37°C and in 5% $CO_2$. BCBL-1 (body cavity B lymphoma cells with KSHV infection) was obtained from the AIDS Research and Reference Reagent Program, Division of AIDS, NIAID, NIH and cultured in RPMI 1640 containing 10% FBS at 37°C and in 5% $CO_2$. All cell lines used in this study were authenticated by STR profiling and confirmed to be free of mycoplasma contamination.

CaSki cells were treated with 20 µM PI3K inhibitor, LY294002 (Cell Signaling Technology, Danvers, MA, USA #9901) or 10 µM mitogen-activated protein kinase kinase (MEK) inhibitor, U0126 (Cell Signaling Technology, Danvers, MA, USA #9903) and examined for their proliferation and viability by CCK-8 assay and RAS effector expression by western blot.

## Raft tissues and cervical tissue sections

Raft tissues with or without HPV16 or HPV18 infection and cervical cancer tissue sections from Zhejiang University Women's Hospital were described in our previous reports (*Liu et al., 2021*; *Xu et al., 2019*).

## siRNAs, plasmid constructions, and transfections

Synthetic double-stranded siRNA targeting the lnc-FANCI-2 transcript is listed in *Supplementary file 5*. Human MAP4K4 siRNA (#M-003971), human IGFBP3 siRNA (#M-004777), and non-targeting control siRNA (#D-001210-01) were purchased from Dharmacon, Inc (Lafayette, CO, USA). Each siRNA transfection into CaSki or other type of cells was carried out by LipoJet in vitro transfection reagent (SignaGen, Frederick, MD, USA). Plasmid pHBL03 has a short isoform of lnc-FANCI-2 cDNA (lnc-FANCI-2a-PA1, GenBank Acc. No. MT669801). Overlapping PCR products generated from 5′ and 3′ RACE PCR products with a primer pair of oHBL 17 and oHBL 18 were digested and inserted into pcDNA3 at Hind III and Xba I sites. The insertion was verified by sequencing. Plasmid pHBL05 has a long isoform of lnc-FANCI-2 (lnc-FANCI-2a-PA2, GenBank Acc. No. MT669800). PCR products generated from genomic DNA with oHBL 19 and oHBL 23 were digested and swapped into pHBL03 to replace lnc-FANCI-2a-PA1 at Bsmb I and Xba I sites.

CRISPR/Cas9 system was used to knock out the lnc-FANCI-2 gene from CaSki cells. Two lnc-FANCI-2 specific gRNAs were designed by an online CRISPR design tool developed by the Zhang lab (https://www.zlab.bio/guide-design-resources). A modified cloning strategy was used to clone the two gRNA into pSpCas9(BB)-2A-Puro vector (Addgene, Watertown, MA, USA #62988) and described in our previous publication (*BeltCappellino et al., 2019*). This strategy allows all three components (two gRNAs and Cas9) to be expressed from the same plasmid to ensure that the transfected cells express two gRNAs simultaneously to knock out the targeted gene (*BeltCappellino et al., 2019*). The sequences of primers to make gRNAs were listed in *Supplementary file 5*.

## Knockout of lnc-FANCI-2 gene in CaSki cells, PCR screening, and genotyping

Two gRNAs each under control by a U6 promoter in a pSpCas9(BB)-2A-Puro vector were used to create deletion in CaSki cells. Plasmid pHBL25, which contains gRNA 1 and 6, was used to delete the entire promoter region of lnc-FANCI-2, and plasmid pHBL27, which contains gRNA 3 and 4, was used to delete only two YY1-binding motifs spanning over an 87 bp region in the lnc-FANCI-2 promoter. CaSki cells transfected with the gRNA-expressing vectors were then under 2 weeks of puromycin selection (0.1 μg/ml). The puromycin-resistant CaSki cells were diluted to a final concentration of 0.5 cells per 100 μl, plated 100 μl of diluted cells into each well of a 96-well plate, and grew under continuous puromycin selection. The colonies were inspected to ensure the clones were grown from one cell and refed or re-plated as needed in 3 weeks of expansion.

A direct PCR was performed for a number of single-cell clones as described (*BeltCappellino et al., 2019*). Briefly, several hundred cells expanded from a single-cell clone were resuspended in 50 μl of phosphate-buffered saline (PBS) and then frozen-thawed on dry ice three times. Cell lysates were treated with 4 μl of QIAGEN protease (QIAGEN, Hilden, Germany, #1017782) at 56°C for 10 min, and the protease was inactivated at 95°C for 5 min. The resulting product was directly used for PCR screening. P1 + P5 were used for verification of promoter deletion and P3 + P4 for YY1 motifs deletion. To ensure the selected single-cell clones displaying homozygous knockout of lnc-FANCI-2, total cell DNA was isolated using a QIAamp DNA Blood Minikit (QIAGEN, Hilden, Germany, #51106) and then the homozygous deletion was screened and the cells with homozygous KO were selected and confirmed by PCR. The primers used for genotyping are listed in *Supplementary file 5*. At least two single-cell clones either with lnc-FANCI-2 promoter deletion or with YY1 motif deletion were selected. Three single-cell clones from the cells transfected with an empty vector, pSpCas9(BB)-2A-Puro, were also selected and served as controls for the genotyping screening.

## Cell proliferation and viability assay

Cell proliferation and viability were determined by a CCK-8 assay (Dojindo Molecular Technologies, Rockville, MD, USA). For the proliferation assay, 500 μl of cell suspension (50,000 cells/well) were dispensed into individual wells in a 24-well plate. Cells were then treated with inhibitors (20 μM LY294002 or 10 μM U0126, respectively) after 24 hr culture. At each time point of 24 hr, 48 hr, and

72 hr, 50 µl of CCK-8 solution was added in three wells of each group and incubated for 1 hr in a cell culture incubator at 37°C. Cell proliferation was determined by absorbance at 450 nm, and normal medium was used to subtract background. Percentage of cell viability was calculated by the following formula: cell viability %=OD450 of inhibitors treated sample/OD450 of untreated sample × 100%.

## Colony formation, cell migration assay, and senescence analysis

CaSki cells were seeded at $1 \times 10^4$, $5 \times 10^3$, or $1 \times 10^3$ cells/well in six-well plates. After 24 hr of cell growth, the culture medium in each well was replaced by DMEM containing 1% methylcellulose and 10% FBS. The plates were fixed with 3.7% formaldehyde and stained with 1% crystal violet in 2 weeks. Transwell cell migration assay was performed as previously described (*Justus et al., 2014*). Briefly, CaSki cells were resuspended in serum-free DMEM containing 0.1% bovine serum albumin (BSA). $1 \times 10^5$ cells in 100 µl were then plated on top of the filter membrane in a transwell insert (SARSTEDT, Nümbrecht, Germany, # 83.3932.800) and incubated for 10 min at 37°C to allow the cells to settle down, then 600 µl of complete media containing 10% FBS were added into the bottom of the lower chamber in a 24-well plate. After 24 hr in culture, the cells on the top of the insert membrane were removed, and cells on the bottom were fixed, stained by 0.2% crystal violet, and then counted.

Cell senescence was determined by Cellular Senescence Assay (Millipore Sigma, Burlington, MA, USA, # QIA117-1KIT). Briefly, CaSki cells were plated at $2 \times 10^5$ cells/well in six-well plates and cultured for 2 days to 50% confluence. Cells were washed once with 2 ml PBS and fixed with 0.5 ml of fixative solution at room temperature for 10–15 min. Then, cells were washed twice with 2 ml PBS, and 1 ml staining solution was added to each well. Cells were incubated at 37°C overnight without $CO_2$ and protected from the light. Blue stained cells were then counted under light microscopy.

## Nuclear and cytoplasmic fractionation

CaSki and SiHa cells were fractionated by using Nuclei EZ Prep Kit (Sigma-Aldrich, #NUC-101) following the manufacturer's protocol. The details of the method could be found in our previous publication (*Yu and Zheng, 2022*). Western blot analysis for nuclear protein SRSF3 and cytoplasmic protein GAPDH was used to determine fractionation efficiency. The fractionated cytoplasmic and nuclear RNAs were used for the detection of lnc-FANCI-2 RNA with human GAPDH RNA serving as a control for RNA fractionation efficiency of the cytoplasmic RNAs.

## RT-qPCR

Detection of lnc-FANCI-2 by RT-qPCR was performed as described (*Liu et al., 2021*). Pre-designed primers for lnc-FANCI-2 are listed in *Supplementary file 5*. Briefly, 2 µg total RNA was converted to cDNA using Superscript First-Strand Synthesis kit (Thermo Fisher Scientific, Waltham, MA, USA #11904018). qPCR was performed using TaqMan gene expression Master Mix (Applied Biosystems, Waltham, MA, USA #4369016) on a StepOne Plus Real-Time PCR system (Applied Biosystems). TaqMan Gene Expression Assay (Applied Biosystems, #4331182) for GAPDH (Hs02758991_g1) was served as an internal control. Data were plotted as fold change over the control group using the $2^{-\Delta\Delta Ct}$ method by which the data was normalized first to the values for GAPDH and then to the median value for control samples. Data are presented as a bar graph with mean ± SD for each group.

## Northern blot

To validate lnc-FANCI-2 KO in CaSki cells, northern blot was performed using polyA+ mRNA isolated from 100 µg of total CaSki RNA with PolyATtract mRNA Isolation Systems (Promega, #Z5310) or 10 µg of total CaSki total RNA as described previously (*Liu et al., 2021*). Briefly, RNA samples were denatured in NorthernMax Formaldehyde loading dye (Thermo Fisher Scientific, #AM8552) at 75°C for 15 min, then separated in a 1% formaldehyde-containing agarose gel and transferred onto a Gene-Screen Plus hybridization transfer membrane (Perkin Elmer, Waltham, MA, USA #NEF987001PK). RNAs on the membrane were cross-linked by exposing to UV light and then prehybridized with PerfectHyb Plus hybridization buffer (Sigma-Aldrich, St. Louis, MO, USA #H7003) for 2 hr at 42°C. Specific oligos against lnc-FANCI-2 or GAPDH listed in *Supplementary file 5* were labeled with [γ-$^{32}$P]-ATP using T4 PNK (Thermo Fisher Scientific, Waltham, MA, USA #18004-010) and added into hybridization buffer for overnight incubation at 42°C. The membrane was then washed with 2× SSPE/0.5% sodium dodecyl

sulfate (SDS) solution for 5 min, followed by two washes with 0.5× SSPE/0.5% SDS each for 15 min at 42°C and then exposed to a PhosphorImager screen.

## RNA-seq and data analysis

Total RNA was extracted from WT CaSki, ΔYY1-D5, and ΔPr-A9 cells using TRIzol reagent (Thermo Fisher Scientific, Waltham, MA, USA #15596018). Total ribo-minus RNA-seq libraries, four samples in each group, were prepared with TruSeq Stranded Total RNA Library Kit and then subjected to paired-end sequencing using Illumina-HiSeq3000/4000 platform.

The obtained reads were processed using the CCBR Pipeliner utility (*CCBR, 2023*). Briefly, reads were trimmed from adapters and low-quality bases using Cutadapt (version 1.18) (https://bioweb. pasteur.fr/packages/pack@cutadapt@1.18) before alignment to the custom reference genome described below. The transcripts were aligned using STAR v2.5.2b in two-pass mode (*Dobin et al., 2013*). Expression levels were quantified using RSEM (RNA-Seq by Expectation-Maximization) (version 1.3.0) (*Li and Dewey, 2011*) with a custom gene annotation described below.

The custom reference genome allowing quantification of both HPV16 and host expression used in this alignment consisted of the human reference genome (hg38/Dec. 2013/GRCh38) with an HPV16 FASTA (https://pave.niaid.nih.gov/) sequence added as an additional pseudochromosome. The custom gene annotation used for gene expression quantification consisted of a concatenation of the hg38 GENCODE annotation version 30 (*Harrow et al., 2012*) and the HPV16 genome, with one other notable alteration. The hg38 v30 gene annotation is incorrect at the location of the lnc-FANCI-2 locus in the hg38 genome at chr15:89378104–89398487, which affects quantification of this gene. To correct this annotation and provide accurate quantification, therefore, we first performed BLAST against hg38 using the sequences of the 14 known isoform transcripts of lnc-FANCI-2 (MT669800 for lnc-FANCI-2a-PA2, MT669801 for lnc-FANCI-2a-PA1, MT669802 for lnc-FANCI-2b-PA2, MT669803 for lnc-FANCI-2b-PA1, MT669804 for lnc-FANCI-2c-PA2, MT669805 for lnc-FANCI-2c-PA1, MT669806 for lnc-FANCI-2d-PA2, MT669807 for lnc-FANCI-2d-PA1, MT669808 for lnc-FANCI-2e-PA2, MT669809 for lnc-FANCI-2e-PA1, MT669810 for lnc-FANCI-2f-PA2, MT669811 for lnc-FANCI-2f-PA1, MT669812 for lnc-FANCI-2g-PA2, and MT669813 for lnc-FANCI-2g-PA1). The results were used to determine the precise exon start and stop sites for each of the isoforms. These were then used to manually adjust the default hg38 v30 gene annotations at the locus indicated above to account for the correct structure of the lnc-FANCI-2 locus and its 14 isoforms.

Downstream analysis and visualization were performed within the NIH Integrated Analysis Portal (NIDAP) using R programs developed on the Foundry platform (Palantir Technologies). Briefly, raw counts data produced by RSEM were imported into the NIDAP platform, genes were filtered for low counts (<1 cpm), and the voom algorithm (*Law et al., 2014*) from the Limma R package (version 3.40.6) (*Smyth, 2004*) was used for quantile normalization and calculation of DEGs. Pre-ranked GSEA was performed using the Molecular Signatures Database version 6.2 (*Liberzon et al., 2011*; *Subramanian et al., 2005*) and the fgsea package (*Korotkevich et al., 2021*). Raw data and the analyzed RNA-seq data supporting the findings in this study have been deposited in the NCBI GEO database (the accession#: GSE190904).

## Human soluble receptor array

To detect the changes of soluble receptors expressed and released from parental WT CaSki cells and lnc-FANCI-2 KO cells, Proteome Profiler Human sReceptor (Soluble Receptor) Array (R&D Systems, Minneapolis, MN, USA # ARY012) was performed according to the manufacturer's protocol. Briefly, 7 × 10⁶ WT CaSki cells and ΔPr-A9 cells were cultured in 15 ml of DMEM supplemented with 10% FBS in a T75 flask for 24 hr. Cell culture supernatants were then collected and centrifuged at 4000 × *g* for 15 min to remove cell debris. Cells were rinsed by 10 ml of PBS and solubilized in lysis buffer 17 at 4°C for 30 min. The cell lysis was then centrifuged at 14,000 × *g* for 5 min to remove cell debris. Protein concentration of cell lysis was measured by Micro BCA Protein Assay (PIERCE, Waltham, MA, USA #23235). Next, two sets of N and C membranes were blocked by Buffer 8/1 in 4-Well Multi-dish for 1 hr on a rocking platform. 500 µl of culture supernatant or 100 µg cell lysate generated from the WT CaSki cells or ΔPr-A9 cells were diluted in 3 ml of Buffer 8/1 and then applied to each membrane after three washes with 1× wash buffer. The membranes were incubated overnight at 4°C on a rocking platform and washed three times to remove unbound materials, followed by incubation with their specific

cocktail of biotinylated detection antibodies for 2 hr at room temperature on a rocking platform. 2 ml of diluted Streptavidin-HRP was then applied to each membrane for 30 min at room temperature on a rocking platform after three times of wash. 1 ml of the prepared Chemi Reagent Mix was added onto the membranes after three washings. The signal was then captured by the ChemiDoc Touch Imaging System (Bio-Rad). Quantification of the relative protein expression in ΔPr-A9 cells compared to the WT cells was determined in Image Lab software (Bio-Rad). The average signal of duplicate spots subtracting background value from negative control spots represented the levels of each protein. The relative change in protein levels was then determined by comparing ΔPr-A9 cells to the WT cells.

## RAS GTPase Chemi ELISA

RAS GTPase activity in cell lysate of the WT CaSki or ΔPr-A9 cells was determined by RAS GTPase Chemi ELISA Kit (Active Motif, Carlsbad, CA, USA # 52097) according to the manufacturer's protocol. Briefly, $2 \times 10^7$ cells were solubilized in 500 µl of Complete Lysis/Binding buffer at 4°C for 15 min. Protein concentration of cell extract was then measured by Micro BCA Protein Assay (PIERCE, #23235) after cell debris was removed by centrifugation at $14,000 \times g$ for 10 min. 2 µg of GST-Raf-RBD in 50 µl of Complete Lysis/Binding buffer was coated in each well of a 96-well plate by incubating for 1 hr at 4°C with 100 rpm agitation. After three washings, 50 µg of extract in 50 µl Complete Lysis/Binding buffer was then added to each well. HeLa (EGF-treated) extract was served as a positive control. The plate was incubated for 1 hr at room temperature with 100 rpm agitation. The active RAS protein (GST-Raf-RBD binding) in each well was then detected by incubating with primary RAS antibody and three times of wash, and then, HRP-conjugated secondary antibody and three times of wash. Chemi-luminescence in each well was read in a luminometer by adding 50 µl room temperature chemilumi-nescent working solution.

## RNA-ISH and IF staining

Endogenous lnc-FANCI-2 in cervical cancer tissues and their derived cell lines was examined by RNAscope Multiplex Fluorescent V2 Assay (Advanced Cell Diagnostics, Minneapolis, MN, USA #323100) as described previously (*Liu et al., 2021*). A custom-designed probe targeting nt 359–1713 of GenBank Acc. No. MT669800.1 transcript for lnc-FANCI-2a-PA2 was utilized. Dual staining of lnc-FANCI-2 and IGFBP RNA was performed according to RNAscope Multiplex Fluorescent v2 Manual part 2 (#323100), and the channel one probe for IGFBP3 (#310351) and channel three probe for lnc-FANCI-2 (#509061-C3) were applied.

For RNA-ISH combined IF staining of MCAM protein, $3 \times 10^5$ of CaSki cells were grown on a glass coverslip in a six-well plate for 24 hr before transfection. Plasmid pHBL05 containing a long isoform of lnc-FANCI-2a-PA2 (GenBank Acc. No. MT669800.1) was transfected into cells with LipoD 293 DNA in vitro transfection reagent. Cultured Adherent Cell Sample Preparation for the RNAscope Multiplex Fluorescent v2 was performed according to the manufacturer's protocol (Advanced Cell Diagnostics, Minneapolis, MN, USA MK-50-010). Briefly, the cells were washed with PBS and fixed by 10% neutral buffered formalin at room temperature for 30 min. The cells were washed with PBS three times and dehydrated by 50% ethanol for 5 min, 70% ethanol for 5 min, 100% ethanol for 5 min, and 100% ethanol for 10 min. The cells were then rehydrated by 70% ethanol for 2 min, 50% ethanol for 2 min, and PBS for 10 min. The cells were then applied to hydrogen peroxide at RT for 10 min and washed with distilled water three times. The cells were then applied to Protease III (1:15 dilution with PBS) at RT for 10 min. The cells were ready for RNAscope Multiplex Fluorescent v2 Manual part 2 (#323100). IF was performed after RNA-ISH but before counterstaining with DAPI. The cells were washed with PBS and were then blocked with 2% BSA in PBS for 1 hr at 37°C or overnight at 4°C. The cells were incubated with the MCAM primary antibody (ProteinTech, Rosemont, IL, USA #17564-1-AP, diluted 1:200 in 2% BSA blocking buffer) for 2 hr at 37°C. An Alexa Fluor-conjugated secondary antibody (1:500, Thermo Fisher Scientific) was diluted in a blocking solution and incubated for 1 hr at 37°C. The slides were washed with PBS three times. DAPI was used for nuclei counterstaining before mounting in a Prolong Gold Antifade mounting medium (Thermo Fisher Scientific, #P36934).

Confocal images were collected using a Zeiss LSM710 laser-scanning microscope equipped with a 63 × Plan-Apochromat (NA 1.4) objective lens. Three-dimensional distributions of lnc-FANCI-2 were generated by Z stacks using ZEN 2.3 software (Zeiss).

## IRPCRP and LC-MS/MS analysis

To identify the lnc-FANCI-2 RNA-associated proteins, we pulled down lnc-FANCI-2 RNA from WT CaSki cells using IRPCRP, a modified ChIRP protocol from the published method (*Chu et al., 2012*), followed by mass spectrometry (*Chu and Chang, 2018*). Briefly, a total of 30 lnc-FANCI-2 anti-sense oligo probes (oLLY496-oLLY525), each with biotinTEG at 3' end, across entire lnc-FANCI-2 RNA (GenBank Acc. No MT669800.1) was designed using the online probe designer (singlemoleculefish.com). Two probe pools were used in IRPCRP assays: the probe pool 1 was the mixed oligos in even numbers (oLLY496/498/500/502/504/506/508/510/512/514/516/518/520/522/524) and the probe pool 2 mixed with the oligos in odd numbers (oLLY497/499/501/503/505/507/509/511/513/515/517/519/521/523/525) (*Supplementary file 5*). CaSki cells were seeded and harvested at 24 hr, and the cell lysates from 200 million cells were used for each IRPCRP reaction. Instead of formaldehyde cross-link in the standard protocol (*Chu et al., 2012*), UV cross-linking (254 nm, energy at 480 mJ/cm$^2$) was applied in our study to minimize non-specific binding. After UV cross-linking, the cells were lysed on ice in 1 × radioimmunoprecipitation assay buffer (Boston Bioproducts, Ashland, MA, USA #BP-115) (50 mM Tris base/Tris-HCl [pH 7.4], 150 mM NaCl, 0.5% sodium deoxycholate, 0.1% SDS, 1% NP-40) supplemented with protease inhibitors (complete mini-EDTA-free protease inhibitor cocktail, Millipore Sigma, Burlington, MA, USA #469315900) and RNase inhibitor (Thermo Fisher Scientific, #AM2694) for 30 min, followed by a brief sonication. The obtained cell lysates were spun for 10 min of centrifugation at 20,000 × *g* at 4°C, and the collected cell lysate supernatants were pre-absorbed with C-1 magnetic beads (Thermo Fisher Scientific, #65002) for 1 hr at room temperature before proceeding to oligo probe hybridization.

Two different hybridization protocols were used to pull down lnc-FANCI-2 RNA. One had the pre-absorbed cell lysate incubated with two separate probe pools in hybridization buffer (750 mM NaCl, 0.1% SDS, 50 mM Tris-Cl pH 7.0, 1 mM EDTA, 15% formamide) at room temperature overnight, and the pre-washed C-1 magnetic beads were then added and incubated for an additional 4 hr; the other had two separate oligo probe pools immobilized onto pre-washed C-1 magnetic beads first at room temperature for 1 hr and then mixed with the pre-absorbed cell lysates in hybridization buffer overnight at room temperature, along with the negative control-beads only IRPCRP reaction without oligo probes serving as a control in the modified protocol. After hybridization, the beads were washed with 1 × wash buffer (2 × NaCl and sodium citrate, 0.1% SDS) for five times and resuspended in 1 ml lysis buffer. 100 µl beads from each IRPCRP were set aside for RNA isolation of lnc-FANCI-2 pull-down efficiency, and the leftover beads were sent for mass spectrometry analysis. The input and IRPCRP RNA were isolated using a MIRNeasy mini kit (QIAGEN, #217004) following the standard protocol after proteinase K (Millipore Sigma, #71049) digestion.

For mass spectrometry analysis, the IRPCRP beads were processed for trypsin/LysC digestion before being submitted to Thermo Scientific Orbitrap Exploris 240 Mass Spectrometer and a Thermo Dionex UltiMate 3000 RSLCnano System for Proteinomics. Peptides from trypsin digestion were loaded onto a peptide trap cartridge at a flow rate of 5 µl/min. The trapped peptides were eluted onto a reversed-phase Easy-Spray Column PepMap RSLC, C18, 2 µM, 100 A, 75 µm × 250 mm (Thermo Scientific) using a linear gradient of acetonitrile (3–36%) in 0.1% formic acid. The elution duration was 110 min at a flow rate of 0.3 µl/min. Eluted peptides from the Easy-Spray column were ionized and sprayed into the mass spectrometer, using a Nano Easy-Spray Ion Source (Thermo Fisher Scientific) under the following settings: spray voltage, 1.6 kV, Capillary temperature, 275°C. Other settings were empirically determined. Raw data files were searched against human protein sequences database using the Proteome Discoverer 2.4 software (Thermo Fisher Scientific, San Jose, CA, USA) based on the SEQUEST algorithm. All the protein peptides identified in the IRPCRP pull-downs were summarized in *Supplementary file 4*.

## Acknowledgements

We thank Louise T Chow of the University of Alabama at Birmingham for her critical reading of the manuscript. We thank Craig Meyers of Penn State University Hershey Medical Center for providing HPV16- and HPV18-infected raft cultures, Xing Xie and Yang Li of Zhejiang University Women's Hospital of China for cervical tissue sections, and Johannes G Schweizer from Arbor Vita Corporation for anti-HPV16 E6-specific antibodies. This work was supported by the Intramural Research Program of the National Institutes of Health, the National Cancer Institute, and the Center for Cancer Research (ZIA SC 010357 to ZMZ).

## Additional information

### Funding

| Funder | Grant reference number | Author |
|---|---|---|
| The Intramural Research Program of the NIH, NCI, Center for Cancer Research | ZIA SC 010357 | Zhi-Ming Zheng |

The funders had no role in study design, data collection and interpretation, or the decision to submit the work for publication.

### Author contributions

Haibin Liu, Conceptualization, Data curation, Software, Formal analysis, Validation, Investigation, Visualization, Methodology, Writing – original draft, Writing – review and editing; Lulu Yu, Data curation, Formal analysis, Validation, Visualization, Methodology, Writing – original draft, Writing – review and editing; Vladimir Majerciak, Conceptualization, Visualization, Writing – review and editing; Thomas J Meyer, Software, Visualization, Methodology; Ming Yi, Software, Investigation, Writing – review and editing; Peter F Johnson, Conceptualization, Methodology; Maggie Cam, Software, Formal analysis; Douglas R Lowy, Supervision, Writing – review and editing; Zhi-Ming Zheng, Conceptualization, Resources, Data curation, Formal analysis, Supervision, Funding acquisition, Investigation, Writing – original draft, Project administration, Writing – review and editing

### Author ORCIDs

Haibin Liu ⓘ https://orcid.org/0000-0003-3601-6066
Lulu Yu ⓘ https://orcid.org/0000-0003-0309-9520
Zhi-Ming Zheng ⓘ https://orcid.org/0000-0001-5547-7912

### Ethics

All clinical cervical tissues were obtained from anonymized excess cervical tissues, with informed consent, that were no longer needed for diagnosis or clinical purposes from a study approved for clinical research by the Institutional Review Board of the Zhejiang University Women's Hospital.

Reviewer #1 (Public review): https://doi.org/10.7554/eLife.102681.3.sa1
Reviewer #3 (Public review): https://doi.org/10.7554/eLife.102681.3.sa2
Author response https://doi.org/10.7554/eLife.102681.3.sa3

---

## Additional files

### Supplementary files

Supplementary file 1. Genome-wide effect of lnc-FANCI-2 knockout (KO) in ΔPr-A9 and ΔYY1 cells on gene expression when compared with parental wild-type (WT) CaSki cells.

Supplementary file 2. The genes with altered expression found in both ΔPr-A9 and ΔYY1-D5 cells when compared with parental wild-type (WT) CaSki cells.

Supplementary file 3. The genes with increased expression in KRAS signaling and epithelial-mesenchymal transition (EMT) and with decreased expression in interferon gamma (IFN-γ) and interferon alpha (IFN-α) responses in lnc-FANCI-2 knockout (KO) cells (ΔPr-A9 and ΔYY1-D5) over the parental wild-type (WT) CaSki cells.

Supplementary file 4. CaSki proteins and their peptide spectrum matches (PSMs) identified by lnc-FANCI-2 Isolation of lnc-FANCI-2 RNA-protein complex by RNA purification (IRPCRP)-LC-MS/MS.

Supplementary file 5. Sequences of oligonucleotides used in this study.

MDAR checklist

### Data availability

RNA Sequencing data have been deposited in GEO under accession code GSE190904.

The following dataset was generated:

| Author(s) | Year | Dataset title | Dataset URL | Database and Identifier |
|---|---|---|---|---|
| Zheng Z, Liu H, Majerciak V, Meyer T | 2022 | Effect of a long non-coding RNA, lnc-FANCI-2 on host transcriptome | https://www.ncbi.nlm.nih.gov/geo/query/acc.cgi?acc=GSE190904 | NCBI Gene Expression Omnibus, GSE190904 |

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
