## [Editor Report · eLife Assessment]

This study reports **important** new insights into the roles of a long noncoding RNA, lnc-FANCI-2, in the progression of cervical cancer induced by a type of human papillomavirus. Through a blend of cell biological, biochemical, and genetic analyses of RNA and protein expression, protein-protein interaction, cell signaling, and cell morphology, the authors provide **convincing** evidence that lnc-FANCI-2 affects cervical cancer outcome by regulating the RAS signaling pathway. These findings will be of interest to scientists in the fields of cervical cancer, long noncoding RNA, and cell signaling.

---

## [Referee Report · Reviewer #1 (Public review)]

Summary:

The authors attempted to dissect the function of a long non-coding RNA, lnc-FANCI-2, in cervical cancer. They profiled lnc-FANCI-2 in different cell lines and tissues, generated knockout cell lines, and characterized the gene using multiple assays.

Strengths:

A large body of experimental data has been presented and can serve as a useful resource for the scientific community, including transcriptomics and proteomics datasets. The reported results also span different parts of the regulatory network and open up multiple avenues for future research.

Weaknesses:

The write-up is somewhat unfocused and lacks deep mechanistic insights in some places.

Comments on revisions:

The manuscript is much improved. I am satisfied with the authors' responses.

---

## [Referee Report · Reviewer #3 (Public review)]

Summary:

A long noncoding RNA, lnc-FANCI-2, was reported to be regulated by HPV E7 oncoprotein and a cell transcription factor, YY1 by this group. The current study focuses on the function of lnc-FANCI-2 in HPV-16 positive cervical cancer is to intrinsically regulate RAS signaling, thereby facilitating our further understanding additional cellular alterations during HPV oncogenesis. Authors used the advanced technical approaches such as KO, transcriptome and (IRPCRP) and LC- MS/MS analyses in the current study and concluded that KO Inc-FANCI-2 significantly increase RAS signaling, especially phosphorylation of Akt and Erk1/2.

Strengths:

(1) HPV E6E7 are required for full immortalization and maintenance of malignant phenotype of cervical cancer, but they are NOT sufficient for full transformation and tumorigenesis. This study helps further the understanding of other cellular alterations in HPV oncogenesis.

(2) lnc-FANCI-2 is upregulated in cervical lesion progression from CIN1, CIN2-3 to cervical cancer, cancer cell lines and HPV transduced cell lines.

(3) Viral E7 of high-risk HPVs and host transcription factor YY1 are two major factors promoting lnc-FANCI-2 expression.

(4) Proteomic profiling of cytosolic and secreted proteins showed inhibition of MCAM, PODXL2 and ECM1 and increased levels of ADAM8 and TIMP2 in KO cells.

(5) RNA-seq analyses revealed that KO cells exhibited significantly increased RAS signaling but decreased IFN pathways.

(6) Increased phosphorylated Akt and Erk1/2, IGFBP3, MCAM, VIM, and CCND2 (cyclin D2) and decreased RAC3 were observed in KO cells.

Comments on revisions:

The revised manuscript has been significantly improved. The authors addressed all my concerns.

---

## [Author Response]

The following is the authors’ response to the previous reviews

**Public Reviews:**

**Reviewer #1 (Public review):**
Summary:The authors attempted to dissect the function of a long non-coding RNA, lnc-FANCI-2, in cervical cancer. They profiled lnc-FANCI-2 in different cell lines and tissues, generated knockout cell lines, and characterized the gene using multiple assays.Strengths:A large body of experimental data has been presented and can serve as a useful resource for the scientific community, including transcriptomics and proteomics datasets. The reported results also span different parts of the regulatory network and open up multiple avenues for future research.

Thanks for your positive comments on the strengths.

Weaknesses:The write-up is somewhat unfocused and lacks deep mechanistic insights in some places.

As the lnc-FANCI-2 as a novel lncRNA had never been explored for any functional study, our report found that it regulates RAS signaling. Thus, this report focuses on lnc-FANCI-2 and RAS signaling pathway but also includes some important screening data, which are important for our readers to understand how we could reach the RAS signaling.

**Reviewer #2 (Public review):**
The study by Liu et al provides a functional analysis of lnc-FANCI-2 in cervical carcinogenesis, building on their previous discovery of FANCI-2 being upregulated in cervical cancer by HPV E7.The authors conducted a comprehensive investigation by knocking out (KO) FANCI-2 in CaSki cells and assessing viral gene expression, cellular morphology, altered protein expression and secretion, altered RNA expression through RNA sequencing (verification of which by RT-PCR is well appreciated), protein binding, etc. Verification experiments by RT-PCR, western blot, etc are notable strengths of the study.The KO and KD were related to increased Ras signaling and EMT and reduced IFN-y/a responses.

Thanks for your positive comments. It did take us a few years to reach this scientific point for understanding of lnc-FANCI-2 function.

Although the large amount of data is well acknowledged, it is a limitation that most data come from CaSki cells, in which FANCI-2 localization is different from SiHa cells and cancer tissues (Figure 1). The cytoplasmic versus nuclear localization is somewhat puzzling.

Regarding lnc-FANCI-2 localization, it could be both cytoplasmic and nuclear in cervical cancer tissues, HPV16 or HPV18 infected keratinocytes, and HPV16+ cervical cancer cell line CaSki cells which contain multiple integrated HPV16 DNA copies. But surprisingly, it is most detectable in the nucleus in HPV16+ SiHa cells which contain only one copy of integrated HPV16 DNA (Yu, L., et al. mBio 15: e00729-24, 2024). No matter what, knockdown of lnc-FANCI-2 expression from SiHa cells induces RAS signaling leading to an increase in the expression of p-AKT and p-Erk1/2 (suppl. Fig. S6B).

**Reviewer #3 (Public review):**
Summary:A long noncoding RNA, lnc-FANCI-2, was reported to be regulated by HPV E7 oncoprotein and a cell transcription factor, YY1 by this group. The current study focuses on the function of lnc-FANCI-2 in HPV-16 positive cervical cancer is to intrinsically regulate RAS signaling, thereby facilitating our further understanding of additional cellular alterations during HPV oncogenesis. The authors used advanced technical approaches such as KO, transcriptome and (IRPCRP) and LC- MS/MS analyses in the current study and concluded that KO Inc-FANCI-2 significantly increases RAS signaling, especially phosphorylation of Akt and Erk1/2.Strengths:(1) HPV E6E7 are required for full immortalization and maintenance of the malignant phenotype of cervical cancer, but they are NOT sufficient for full transformation and tumorigenesis. This study helps further understanding of other cellular alterations in HPV oncogenesis.(2) lnc-FANCI-2 is upregulated in cervical lesion progression from CIN1, CIN2-3 to cervical cancer, cancer cell lines, and HPV transduced cell lines.(3) Viral E7 of high-risk HPVs and host transcription factor YY1 are two major factors promoting lnc-FANCI-2 expression.(4) Proteomic profiling of cytosolic and secreted proteins showed inhibition of MCAM, PODXL2, and ECM1 and increased levels of ADAM8 and TIMP2 in KO cells.(5) RNA-seq analyses revealed that KO cells exhibited significantly increased RAS signaling but decreased IFN pathways.(6) Increased phosphorylated Akt and Erk1/2, IGFBP3, MCAM, VIM, and CCND2 (cyclin D2) and decreased RAC3 were observed in KO cells.

Thanks for your positive comments. It has taken us almost nine years to reach this point to gradually understand lnc-FANCI-2 functions, which are more complex than our initial thoughts.

Weaknesses:(1) The authors observed the increased Inc-FANCI-2 in HPV 16 and 18 transduced cells, and other cervical cancer tissues as well, HPV-18 positive HeLa cells exhibited different expressions of Inc-FANCI-2.

Both HPV16 and HPV18 infections induce lnc-FANCI-2 expression in keratinocytes (Liu H., et al. PNAS, 2021). However, HPV18+ cervical cancer cell lines HeLa and C4II cells (Figure S1A and S1B) do not express lnc-FANCI-2 as we see in HPV-negative cell lines such as HCT116, HEK293, HaCaT, and BCBL1 cells. Although we don’t know why, our preliminary data show that the lnc-FANCI-2 promoter functions well and is sensitive to YY1 binding in lnc-FANCI-2 expressing CaSki and C33A cells in our dual luciferase assays but is much less sensitive to YY1 binding in HeLa and HCT116 cells, indicating some unknown cellular factors negatively regulating lnc-FANCI-2 promoter activity.

**Author response image 1. sa3fig1:** A firefly luciferase (FLuc) reporter containing either the wild-type (−600 wt) or YY1-binding-site-mutated lnc-FANCI-2 promoter was evaluated in CaSki, HeLa, C33A, and HCT116 cells for its promoter activity, with Renilla luciferase (RLuc) activity driven by a TK promoter serving as an internal control. The two YY1-binding motifs (A and B) with a X for mutation are illustrated in the right diagram.

(2) Previous studies and data in the current showed a steadily increased Inc-FANCI-2 during cancer progression, however, the authors did not observe significant changes in cell behaviors (both morphology and proliferation) in KO Inc-FANCI-2.

Thanks. We do see decreases in cell proliferation, colony formation, and cell migration, accompanied by increased cell senescence, from the lnc-FANCI-2 KO cells to the parent WT cells. These data are now added to the revised Fig. 1 and the revised supplemental Fig. S3.

(3) The authors observed the significant changes of RAS signaling (downstream) in KO cells, but they provided limited interpretations of how these results contributed to full transformation or tumorigenesis in HPV-positive cancer.

As we stated in the title of this function of lnc-FANCI-2, the lnc-FANCI-2 intrinsically restricts RAS signaling and phosphorylation of Akt and Erk in HPV16-infected cervical cancer. Presumably, high RAS-AKT-ERK signaling inhibits tumor cell survival due to senescence induction as we show in our new Figure 1 and supplemental Fig. S3. A similar report was found in a lung cancer study (Patricia Nieto, et al. Nature 548: 239-243, 2017).

**Recommendations for the authors:**

**Reviewer #1 (Recommendations for the authors):**
Major comments:(1) A major issue is that parts of the manuscript read like a collection of experimental results. However, some of the results do not contribute directly to the central story. Besides confusing the reader, the large amount of apparently disparate results can raise more questions. For example:a) Why is lnc-FANCI-2 highly expressed in HPV16-infected cervical cancer cell lines (but not in HPV18-infected cells)?b) How do p53 and RB repress the expression of lnc-FANCI-2?c) What regulates the sub-cellular localization of lnc-FANCI-2?d) How does lnc-FANCI-2 negatively regulate RAS signalling?e) How does MAP4K4 bind to lnc-FANCI-2?f) Do lnc-FANCI-2 and MAP4K4 require each other to regulate RAS signalling?g) How does RAS signalling regulate the transcription of MCAM and IGFBP3?h) How does MCAM feedback on RAS? Do the different MCAM isoforms impact on RAS signalling differently?i) How does IGFBP3 feedback on ERK but not AKT?j) How do the other mentioned proteins like ADAM8 fit into the regulatory network?k) Each question will require a lot more work to address. I think it would be good if the authors could think through carefully what the key message(s) in the current manuscript should be and then present a more focused write-up.

Thanks for the critical comments. Because this study is the first time to explore lnc-FANCI-2 functions, we would like to be collective. We believe these data are important to guide any future studies. We really appreciate our reviewer listing many questions related to HPV infection, cell biology, RAS signaling, cancer biology from questions a to k. To address each question in a satisfactory way will be a separate study, but fortunately, our report has pointed out such a direction with some preliminary data for future studies. Here below are our responses to each question from a to k:

a) Both HPV16 and HPV18 infection induce lnc-FANCI-2 expression in keratinocytes (Liu H., et al. PNAS, 2021). However, HPV18+ cervical cancer cell lines HeLa and C4II cells (Figure S1A and S1B) do not express lnc-FANCI-2 as we see in HPV-negative cell lines such as HCT116, HEK293, HaCaT, and BCBL1 cells. Although we don’t know why, our preliminary data show that lnc-FANCI-2 promoter functions well and is sensitive to YY1 binding in lnc-FANCI-2 expressing CaSki and C33A cells but is much less sensitive to YY1 in HeLa and HCT116 cells, indicating some unknown cellular factors negatively regulating lnc-FANCI-2 promoter activity.

b) We don’t know whether p53 and pRB could repress the expression of lnc-FANCI-2 although C33A cells bearing a mutant p53 and mutant pRB express high amount of lnc-FANCI-2. However, KD of E2F1 had no effect on lnc-FANCI-2 promoter activity in CaSki cells (Liu, H., et al. PNAS, 2021).

c) RNA cellular localization can be affected by many factors, including splicing, export, and polyadenylation. As lnc-FANCI-2 is a long non-coding RNA, its regulation of cellular location could be more complicated than mRNAs and thus could be a future research direction.

d) The conclusion that lnc-FANCI-2 negatively regulates RAS signaling is based on both lnc-FANCI-2 KO and KD studies. Please see the proposed hypothetic model in Figure 8E.

e) The MAP4K4 binding to lnc-FANCI-2 was demonstrated by our IRPCRP-Mass spectrometry (Fig. 8A and 8C), although the exact binding site on lnc-FANCI-2 was not explored. As you probably know, many enzymes today turn out an RNA-binding enzyme (Castello A., et al. Trends Endocrinol. Metab. 26: 746-757, 2015; Hentze MW., et al. Nat. Rev. Mol. Cell Biol. 19: 327-341, 2018)

f) Yes, they are slightly relied on each other in regulating RAS signaling. We found that KD of MAP4K4 in parent CaSki cells (Figure 8D) led to more effect on RAS signaling (MCAM, IGFBP3, p-Akt) than that in lnc-FANCI-2 KO ΔPr-A9 cells. In contrast, the latter displayed more p-Erk1/2 than that induced by KD of lnc-FANCI-2 in the parental CaSki cells (Figure S7C).

g) We believe RAS signaling regulates most likely the transcription of MCAM and IGFBP3 through phosphorylated transcription factors (Figure 8E diagram).

h) As a signal molecule with at least 13 ligands/coreceptors (Joshkon A., et al. Biomedicines 8: 633, 2020), the increased MCAM appears to sustain RAS signaling (Fig. 7J and Fig. 8E). We are assuming the full-length cytoplasmic MCAM plays a predominant role in RAS signaling due to its abundance than the cleaved nuclear MCAM missing both transmembrane and cytoplasmic regions. Plus, RAS signaling mainly occurs in the cytosol.

i) Exact mechanism remains unknown. Lnc-FANCI-2 KO cells exhibit high expression levels of IGFBP3 RNA and protein and p-Erk1/2, but not so much for p-Akt, possibly due to IGFBP3 regulation of MAPK for Erk phosphorylation, but not much so on PI3K for Akt phosphorylation.

j) The dysregulation of RAS signaling and ADAM protein activity is implicated in various cancers. ADAM proteins can modulate RAS signaling by cleaving and releasing ligands that activate or inactivate RAS-related pathways (Schafer B., et al. JBC 279: 47929-38, 2004; Ohtsu H., et al. Am J Physiol Cell Physiol 291: C1-C10, 2006; Dang M, et al. JBC 286: 17704-17713, 2011; Kleino I, et al. PLoS One 10: e0121301, 2015). Some ADAM proteins are Involved in the migration and invasion of cancer cells, and its loss can promote the degradation of KRAS (Huang Y-K., et al. Nat Cancer 5: 400-419, 2024). In this revision, we have a brief discussion on ADAMs and RAS signaling.

k) We agree with our reviewer that each question will require a lot more work to address. As this study is to explore the lnc-FANCI-2 function for the first time, however, we prefer to include all of these data that have been selectively included in this write-up. We hope reviewer 1 will be satisfied with our response to each question from a to j.

(2) Figures S1A & S1C - Replicates are needed.

Yes, we have repeated all of the experiments. The quantification shown in Figure S1A and S1C was performed in triplicate, and error bars have been added to the updated figure.

1. Figure S1D - There seems to be some lnc-FANCI-2 RNA in the nucleus of CaSki cells as well. Please quantify the relative amount of lnc-FANCI-2 in the nucleus vs cytoplasm.

Yes, a small fraction of lnc-FANCI-2 is in the nucleus of CaSki cells as we reported (Liu H., PNAS, 2021, Movies S1 and S2). We did quantify by fractionation and RT-qPCR the relative amount of lnc-FANCI-2 in the nucleus vs cytoplasm in Figure S1C.

(4) Figure S2B - (a) For ΔPr-A9 cells, it looks like there is an increase in E6 and a decrease in E7, instead of "little change" as the authors claimed. (b) I suggest checking the protein levels for all the control and KO clones.

Thanks for the questions. We had some variation in E6 and E7 detection and the submitted one was one representative. We grew again the lnc-FANCI-2 KO clones A9 and B3 and reexamined the expression of HPV16 E6/E7 proteins and their downstream targets, p53 and E2F1. As shown in new Figure S3A expt II, we saw again some variations in the detections (~20-30%) and these variations do not reflect a noticeable change for their downstream targets. Thus, we do not consider these changes significantly enough to draw a conclusion in our study, but rather most likely from sampling in the assays.

(5) In the Proteome Profiler Human sReceptor Array analysis, multiple proteins were highlighted as having at least 30% change. But it is unclear how they relate to RAS signaling.

Thanks for this comment. Cellular soluble receptors are essential for RAS signaling, EMT pathway and IFN responses. For example, the dysregulation of RAS signaling and ADAM protein activity is implicated in various cancers. ADAM proteins can modulate RAS signaling by cleaving and releasing ligands that activate or inactivate RAS-related pathways (Schafer B., et al. JBC 279: 47929-38, 2004; Ohtsu H., et al. Am J Physiol Cell Physiol 291: C1-C10, 2006; Dang M, et al. JBC 286: 17704-17713, 2011; Kleino I, et al. PLoS One 10: e0121301, 2015). Some ADAM proteins are Involved in the migration and invasion of cancer cells, and its loss can promote the degradation of KRAS (Huang Y-K., et al. Nat Cancer 5: 400-419, 2024). In this revision, we have a brief discussion on ADAMs and RAS signaling.

(6) Does knockdown of MAP4K4 lead to an increase in MCAM and IGFBP3?

Yes, the MAP4K4 KD from parental WT CaSki cells does lead an increase in MCAM (~70%) and IGFBP3 (~30%) which is like the knockdown of lnc-FANCI-2 shown in the revised Figure 8D.

Minor comments:(7) In the opinion of this reviewer the title is somewhat unwieldy.

Thanks. We have shortened the title as “The lnc-FANCI-2 intrinsically restricts RAS signaling in HPV16-infected cervical cancer”

(8) The abstract can be more focused and doesn't have to mention so many gene names. In fact, the significance paragraph works better as an abstract. For the significance, the authors can provide another write-up on the implications of their research instead.

Thanks. We have revised the abstract and added the implications of this research.

(9) The last sentence of the introduction feels a little abrupt. It would be good to elaborate a little more on the key findings.

Thanks for this critical comment. We have revised as in the following: In this report, we demonstrate that lnc-FANCI-2 in HPV16-infected cells controls RAS signaling by interaction with MAP4K4 and other RNA-binding proteins. Ablation of lnc-FANCI-2 in the cells promotes RAS signaling and phosphorylation of Akt and Erk. High levels of lnc-FANCI-2 and low level of MCAM expression in cervical cancer patients correlate with improved survival, indicating that lnc-FANCI-2 plays a critical role in regulating RAS signaling to affect cervical cancer progression and patient outcomes.

(10) Typo on line 191: Should be ADAM8 and not ADMA8.

Corrected.

**Reviewer #2 (Recommendations for the authors):**
The paper contains a vast amount of data and would greatly benefit from an expanded version of the schematic of Figure 8E summarizing the main results. Including additional details on FANCI-2 regulation by HPV (primarily from previous studies) and its implications for HPV16-driven carcinogenesis would provide a more comprehensive overview.

Thanks for the suggestion. We have modified our Figure 8E to include HR-HPV E7 and YY1 in regulation of lnc-FANCI-2 transcription.

Further specific comments:(1) The introduction may be shortened to increase readability (e.g. lines 77-90; 94-105).

We have shortened the introduction by deletion of the lines 94-105 from our initial submission.

(2) Lines 55-57 the number of cervical cancer diagnoses and mortality need to be updated to the latest literature. The reference is from 2012.

Thanks. We have revised and updated accordingly with a new citation (Bray F., et al: Global cancer statistics 2022: GLOBOCAN estimates of incidence and mortality worldwide for 36 cancers in 185 countries. CA Cancer J Clin 74, 229-263 (2024))

(3) Line 61: Progression rate of CIN3 is incorrect (31% in 30 years according to reference 5).

Thanks. Corrected.

(4) Lines 108-112 are difficult to understand and should be rewritten.

Thanks. Revised accordingly.

(5) Line 116 Is this correct or should 'but' be 'and'?

Thanks. Corrected accordingly.

(6) Figure 1A top: The difference between cervical cancer and normal areas is hard to see in the top figure. The region labeled as "normal" does not resemble typical differentiating epithelium or normal glandular epithelium, though this is difficult to assess accurately from the image provided. I suggest adding HE staining and also the histotypes.

We have added an H&E staining panel in the corresponding region to Figure 1A, which clearly shows the normal and cancer regions. Both cervical cancer tissues were cervical squamous cell carcinoma.

(7) HFK-HPV16 & 18 cells (Figure 1B) are not described in the Materials & Methods.

Thanks. We revised our Materials and Methods by citing our two previous publications.

(8) Figure 2E (RNA scope on FANCI-2 KO) only shows 2 to 3 cells, which makes it somewhat difficult to assess downregulated expression in the KO. I suggest replacing these with pictures showing more cells (i.e. >10) to strengthen the results.

We have replaced the image in Figure 2E to include more cells.

(9) The spindle-like morphology in deltaPr-A9 cells shown in FigS2A is not very distinct. Including images at higher magnification could help clarify this feature.

Good comment. We have enlarged the images for better view and revised the context.

(10) Both protein and RNA expression analysis have been performed on WT CaSki cells and FANCI-2 KO cells. If I am correct there is little overlap between the significantly changed gene products. What does this mean? Have you looked into the comparison?

The DEGs identified from RNA-seq indicated a genome wide transcriptome change, while the protein array we used only covered 105 soluble protein receptors. However, we did find 9/15 (60%) membrane proteins in cell lysates (PODXL2, ECM1, NECTIN2, MCAM, ADAM9, CDH5, ADAM10, ITGA5, NOTCH1, SCARF2, ADAM8, TIMP2, LGALS3BP, CDH13, and ITGB6) exhibited consistent changes in expression (underlined) by both RNA-seq and protein array assays. We have revised the text with this information (page 11). Other six proteins (40%) had inconsistent expression correlation in two assays could be due to post-translational mechanisms, such as protein stability, modifications and secretion, etc.

(11) Figure S7, which represents TCGA data and survival is quite complex. It would be more effective to display a similar figure for FANCI-2, as was done for MCAM in Figure 7I, to simplify the comparison and enhance clarity.

Thanks. However, the suggested figure for lnc-FANCI-2 was published in PNAS paper already (Liu H., et al. PNAS, 2021). The Figure S8 in this revision is the result from our in-house GradientScanSurv pipeline, a new way to correlate the expression and survival more accurately.

What do the Figures look like if you analyse only HPV16+ patients versus HPV18+ patients, considering that FANCI-2 upregulation in cell lines is related to HPV16 and not 18? Is there an effect of histotype? Or tumor stage?

HPV18 infected keratinocytes express high level of lnc-FANCI-2. Two HPV18^+^ HeLa and C4II cell lines and HPV-negative cell lines, such as HCT116 cells, which do not express lnc-FANCI-2 could be due to the presence of some unknow repressive factors. We found that lnc-FANCI-2 promoter functions well in responding to YY1 binding in CaSki and C33A cells expressing lnc-FANCI-2 but does not so in HeLa and HCT116 cells in our dual luciferase assays.

(12) It remains puzzling that FANCI-2 upregulation was previously shown to already occur in CIN lesions and increase further in cervical cancer, while the current data indicate that FANCI-2 suppresses AKT activation. If I am correct Akt activation has been linked to cervical carcinogenesis. Similarly, line 434 states that increased MCAM might promote cervical tumorigenesis, implying that low FANCI-2 would stimulate tumorigenesis. If I understand correctly, the increase in FANCI-2 observed in CIN lesions would reflect a "brake" on the carcinogenic pathway and its sustained increase in cancer might indicate that growth is still (partly) controlled. As mentioned earlier, a Figure illustrating the relation between FANCI-2, HPV, and the carcinogenic process would be beneficial for clarity.

Yes. Increased MCAM, but low level of lnc-FANCI-2, correlates with poor cervical cancer survival. We have revised Figure 8E to illustrate this relation better.

(13) May part of the potentially conflicting findings be explained by CaSki cells being of metastatic origin? Related to this, does the expression of FANCI-2 or MALM depend on the tumor stage?

Thanks for this important suggestion. Unfortunately, we found that the expression of lnc-FANCI-2 and MCAM is not associated with cervical cancer stage based on the TCGA data (http://gepia.cancer-pku.cn/index.html). See the data below:

**Author response image 2. sa3fig2:** 

Despite some lingering uncertainty, the extensive experiments conducted using KO and KD cells do provide compelling evidence that lnc-FANCI-2 function is linked to RAS signaling and EMT.

Thanks for your positive review and instructive comments.

**Reviewer #3 (Recommendations for the authors):**
(1) The authors observed the increased Inc-FANCI-2 in HPV 16 and 18 transduced cells, and other cervical cancer tissues as well, HPV-18 positive HeLa cells exhibited different expressions of Inc-FANCI-2. I suggest authors provide more discussions on this difference, for example, HPV genotypes. HPV genome status in host cells? Cell types?

Thanks. We found the keratinocyte infections with HPV16, HPV18, and other HR-HPVs could induce lnc-FANCI-2 expression (Liu H., et al. PNAS, 2021). In this report, we found HPV18^+^ HeLa and C4II cells and other HPV-negative cell lines do not. Our preliminary data on lnc-FANCI-2 promoter activity assays showed the presence of a negative regulatory factor (s) in non-lnc-FANCI-2 expressing cells. See the data in Author response image 1.

We have revised our discussion by inclusion these sets of the luciferase data as data not shown.

(2) I suggest the authors discuss more details on how the changes of RAS signaling in KO cells help our further understanding of the molecular mechanisms for HPV-associated full-cell transformation and malignancy in addition to the well-known functions of HPV E6 and E7.

Thanks. We have modified the Figure 8E as suggested by reviewer 2 and revised the discussion further.